# Masked Generative Adversarial Networks are Data-Efficient Generation Learners

**Jiaxing Huang[1]\*, Kaiwen Cui[1]\*, Dayan Guan[1], Aoran Xiao[1], Fangneng Zhan[2],**
**Shijian Lu[1]†, Shengcai Liao[3], Eric Xing[45]**

[1] School of Computer Science and Engineering, Nanyang Technological University, Singapore
[2] Max Planck Institute for Informatics, Germany
[3] Inception Institute of Artificial Intelligence (IIAI), Abu Dhabi, UAE
[4] School of Computer Science, Carnegie Mellon University, USA
[5] Mohamed bin Zayed University of Artificial Intelligence, Abu Dhabi, UAE

`{Jiaxing.Huang, Kaiwen.Cui, Dayan.Guan, Aoran.Xiao, Shijian.Lu}@ntu.edu.sg`
`fzhan@mpi-inf.mpg.de, shengcai.liao@inceptioniai.org`
`epxing@cs.cmu.edu, Eric.Xing@mbzuai.ac.ae`

## Abstract

This paper shows that masked generative adversarial networks (MaskedGAN) are robust image generation learners with limited training data. The idea of MaskedGAN is simple: it randomly masks out certain image information for effective GAN training with limited data. We develop two masking strategies that work along orthogonal dimensions of training images, including a *shifted spatial masking* that masks the images in spatial dimensions with random shifts, and a *balanced spectral masking* that masks certain image spectral bands with self-adaptive probabilities. The two masking strategies complement each other which together encourage more challenging holistic learning from limited training data, ultimately suppressing trivial solutions and failures in GAN training. Albeit simple, extensive experiments show that MaskedGAN achieves superior performance consistently across different network architectures (*e.g.*, CNNs including BigGAN and StyleGAN-v2 and Transformers including TransGAN and GANformer) and datasets (*e.g.*, CIFAR-10, CIFAR-100, ImageNet, 100-shot, AFHQ, FFHQ and Cityscapes).

## 1 Introduction

*"What I cannot create, I do not understand."* —*Richard Feynman.* Generative Adversarial Network (GAN) [19] is such an unsupervised framework learning to understand and create: the discriminator aims to understand "what are real and fake images" by binary discrimination while the generator strives to produce realistic images according to discriminator's feedback. In recent years, GANs have achieved great success under the presence of large amounts of training data. When only limited training data is available, they instead experience obviously increased generation failures [58, 31]. By reversing the dictum, we get *"What I do not understand, I can not create"* [43], which might be the perfect metaphor of such generation failures. Specifically, with limited training data, the discriminator tends to achieve discrimination via certain less meaningful shortcuts by memorizing training data or merely focusing on easy-to-discriminate image locations and spectra without holistic understanding of images, ultimately leading to trivial solutions that fail to "create" realistic images [58, 31] as shown in Fig. 1.

---

\*indicates equal contribution.
†corresponding author.

36th Conference on Neural Information Processing Systems (NeurIPS 2022).

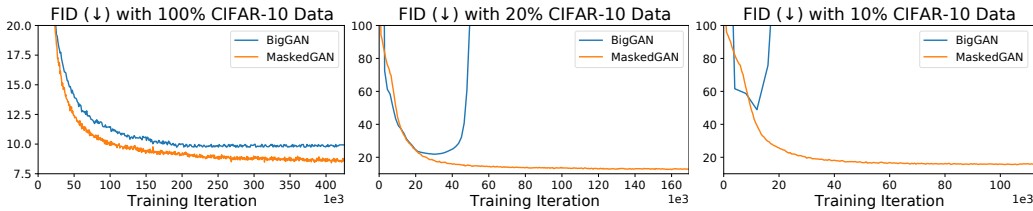

Figure 1: State-of-the-art GANs such as BigGAN often experience clear training collapses with limited training data such as 20% and 10% of CIFAR-10. The proposed MaskedGAN (*i.e.*, Masked-BigGAN) effectively stabilizes adversarial training and suppresses trivial solutions and training failures by encouraging more challenging holistic understanding of images.

Trivial solutions are ubiquitous in unsupervised learning [3, 6, 7, 10], and so in Masked Autoencoders (MAE) [48, 42, 8, 18, 2, 21, 51] as one prevalent unsupervised representation method. MAE works by masking and reconstructing the image, where image masking is the core design that suppresses trivial solutions and enables learning high-level semantic features [21, 5]. For instance, [21] presents a high-ratio masking strategy for MAE: 1) it can impede shortcuts in image reconstruction, *e.g.*, local spatial interpolation that recovers a masked patch from adjacent patches without high-level understanding [21, 5]; 2) it encourages global spatial interpolation that requires modelling inter-patch context information [54, 25] globally, ultimately learning nontrivial and meaningful representations [21, 5].

In this work, we explore the idea of image masking training for GAN, aiming to develop robust image generation learners with limited training data. By viewing image masking as one general means of suppressing training shortcuts, we hypothesize that image masking in GAN training should be conducted over both spatial and spectral dimensions for preventing discriminators from merely focusing on easy-to-discriminate image locations and spectra. In another word, applying random spatial and spectral masking creates more challenging tasks for discriminator training which forces discriminators to learn more holistic instead of easy-to-discriminate information only. At the other end, as applying masking to discriminators slows down their learning, it is crucial to decelerate the generator learning synchronously for an overall balanced and stable adversarial learning.

Driven by this motivation, we design MaskedGAN, a robust image generation network that can learn effectively with limited training data. The idea of MaskedGAN is simple: it randomly masks certain image information to suppress trivial solutions in GAN training. We develop two masking strategies that work along orthogonal dimensions of images. The first is *shifted spatial masking* that masks the image in spatial dimensions with a random shift. Intuitively, random masking along spatial dimensions enforces discriminator to learn over all image locations instead of merely focusing on easy-to-discriminate locations. The second is *balanced spectral masking* that decomposes an image into multiple bands in spectral space and masks a portion of spectral bands with self-adaptive probabilities. Random masking along spectral dimension encourages discriminator to learn hard-to-discriminate bands (*e.g.*, high-frequency bands capturing shapes and structures) instead of largely on easy bands (*e.g.*, low-frequency bands capturing color and brightness). The two masking strategies complement each other which together suppress trivial solutions and training failures by encouraging more challenging holistic discriminator learning from limited training data.

MaskedGAN also applies the two masking strategies to the generator training. Differently, on generator side, the two image masking operations work on outputs by sparsifying the training signals backpropagated from the discriminator. This design slows down the generator learning and stabilizes the overall adversarial learning by keeping similar learning paces for discriminator and generator. It ensures that MaskedGAN can converge to a Local Nash Equilibrium under certain conditions [22].

We summarize the contributions of this work in three aspects. *First*, this paper investigates image masking training for GAN, aiming for building robust image generation learners with limited training data. *Second*, we propose MaskedGAN and design spatial and spectral masking for training robust and effective GAN with limited data. The two masking strategies complement and encourage more challenging holistic understanding of images which help suppress trivial solutions and stabilize the overall adversarial training. *Third*, extensive experiments show that a simple implementation of MaskedGAN achieves superior generation performance consistently across different network architectures and datasets.

## 2 Related Work

This work is related to two main fields of research, namely, generative adversarial networks for image generation and masked autoencoders for unsupervised representation learning.

**Generative Adversarial Network** (GAN) [19] is an unsupervised data synthesis framework that aims to generate a model distribution that mimics a given target distribution. It consists of a generator that generates model distribution and a discriminator that distinguishes the model distribution from the target. In recent years, a series of GANs have been proposed to improve the performance and training stability from different perspectives, such as designing more stable training objectives [1, 36, 20, 37, 45], building better network architectures [44, 38, 39, 55, 33, 4, 28, 26, 14], involving various training strategies [16, 30, 35], etc. Despite the great success obtained, they require large amounts of training data for training effective generation models. When only limited training data is available, they often suffer from clearly increased generation failures with severely degraded performance [58, 31, 27, 9]. In this work, we focus on training effective GANs with limited training data.

**Training GAN with limited data:** Recently, the task of training effective GANs with limited data has attracted increasing attention for alleviating the requirement of large amounts of training data. Most existing efforts address this challenging task from two perspectives. The *first* is *data augmentation* [58, 31, 27] that works via massive hand-crafted data augmentation policies, *e.g.*, [58] employs different types of differentiable augmentation to stabilize GAN training. The *second* is *model regularization* [46, 13, 9] which stabilizes GAN training by regularizing the discriminator training via label flipping [46], network co-training [13], lottery ticket hypothesis [9], etc.

We tackle this challenging task from a perspective of image masking training. Specifically, we design MaskedGAN that exploits two orthogonal image masking strategies for encouraging more challenging holistic understanding of images. Extensive evaluations show that MaskedGAN stabilizes the adversarial training process and suppresses trivial solutions and training failures effectively.

**Masked Autoencoders** (MAE) has greatly advanced the research of unsupervised representation learning recently. Despite different designs, MAE is essentially a denoising autoencoder (DAE) [47] that learns semantic representations by corrupting and reconstructing the input signals. Most existing work can be viewed as a generalized DAE under different types of corruptions, such as masking image pixels, patches, and regions [48, 42, 8, 18, 2, 21], or masking image color channels [57]. For example, Context Encoder [42] masks random image regions of different shapes and learns to reconstruct them. Color Encoder [42] masks image color channels and learns to reconstruct them from grayscale images. Recently, [18, 2, 21] mask random image patches for MAE training. In particular, it is demonstrated in [21] that various image masking strategies can help suppress trivial solutions in MAE training while large mask ratios help learn better representations.

Different from most existing work that explores image masking in MAE training for learning better unsupervised representations, we explore the idea of image masking for GAN training and aim to develop robust image generation learners with limited training data. We design two orthogonal image masking strategies, namely, *shifted spatial masking* and *balanced spectral masking* that help learn robust image generation models effectively, more details to be described in the ensuing subsections.

## 3 Method

### 3.1 Task Definition

This work focuses on the task of training GAN with limited training data. GAN is an unsupervised framework that aims to generate a model distribution that mimics a given target distribution: the generator $G$ learns to generate images $G(z)$ given the latent code $z$ while the discriminator $D$ learns to distinguish the generated images $G(z)$ from the real images $x$. Conventionally, GAN is optimized alternatively with a discriminator loss $\mathcal{L}_d$ and a generator loss $\mathcal{L}_g$ that can be defined as:

$$\mathcal{L}_d = \mathbb{E}_{x \sim p_{\text{data}}} [f_d(-D(x))] + \mathbb{E}_{z \sim p_z} [f_d(D(G(z)))], \tag{1}$$

$$\mathcal{L}_g = \mathbb{E}_{z \sim p_z} [f_g(-D(G(z)))], \tag{2}$$

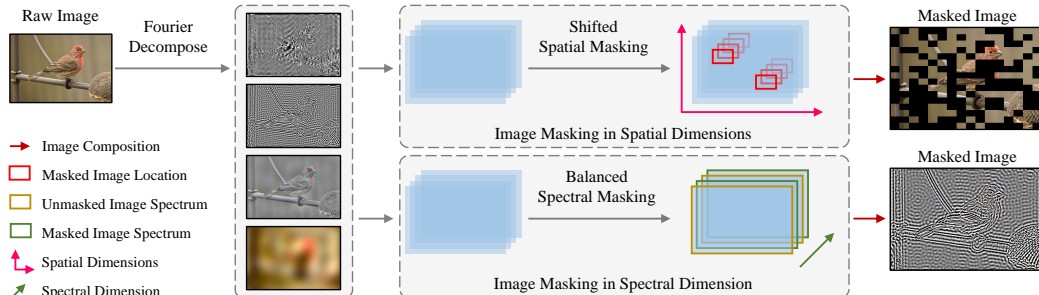

Figure 2: **Overview of the proposed MaskedGAN**: MaskedGAN randomly masks certain image information for effective GAN training with limited training data. We develop two masking strategies that work along orthogonal image dimensions, including *Shifted Spatial Masking* that masks images along spatial dimensions with random shifts and *Balanced Spectra Masking* that masks certain image spectral bands based on self-adaptive probabilities.

where $p_{data}$ denotes the distribution of the provided real images and $p_z$ is the prior distribution (*e.g.*, Gaussian distribution). The notations $f_d$ and $f_g$ stand for the loss mapping functions [19, 38].

**GAN training failures with limited data**. Prior studies [58, 31] show that GAN often experiences generation failures with severely degraded generation performance when only limited training data is available. Specifically, with limited training data, the discriminator tends to discriminate via meaningless shortcuts by merely focusing on easy-to-discriminate image locations and spectra instead of holistic understanding of images. This can be viewed clearly in Fig. 1 of the appendix where the Gini Coefficient [17] of discriminator's spatial attentions increases quickly along the training iteration (when only limited training data is available). Note that the Gini coefficient [17] is negatively correlated with equality, i.e., the discriminator will pay more unevenly distributed attention to each spatial location while the Gini coefficient increases from '0' to '1'. For image generation with GAN, the large Gini coefficient thus means that the discriminator starts to focus on certain spatial locations (easy to discriminate) while ignoring other spatial locations (hard to discriminate), ultimately leading to an over-confident discriminator and training collapse.

### 3.2 Masked Generative Adversarial Network

We tackle the challenging task of training GAN with limited data from a perspective of image masking training [21]. Specifically, we design two orthogonal image masking strategies, namely, *shifted spatial masking* and *balanced spectral masking* as illustrated in Fig. 2. The two masking strategies complement each other and encourage more challenging holistic understanding of images which help stabilize the adversarial training process and suppress trivial solutions and training failures.

**Shifted spatial masking** masks random image patches in spatial dimensions, aiming to force discriminator to learn over all image locations instead of merely focusing on easy-to-discriminate locations. It first generates a random patch-based mask and then introduces a random shift to the generated mask and finally applies the shifted mask to the training image. The shift operation allows to conduct spatial masking continuously along spatial dimensions: a masked patch could appear on any possible locations instead of on fixed grid locations [21, 2]. This feature is critical to image generation that requires high-fidelity along spatial dimensions.

Given an image $x \in \mathbb{R}^{H \times W}$, we first generate a random patch-based spatial mask $m_{\text{spatial}} \in \{0, 1\}^{H \times W}$ and apply a random spatial shift $d \in [0, D]$ on $m_{\text{spatial}}$ to acquire the shifted spatial mask $m'_{\text{spatial}}$. The shifted spatial masking function can be defined by: : $\mathbb{M}_{\text{spatial}}(x) = x \times m'_{\text{spatial}}$. In the patch-based mask generation, we set the grid size as $N \times N$, *i.e.*, the patch size is $\frac{H}{N} \times \frac{W}{N}$. We set the random shift range $D$ as half of patch size, which allows the masked patch to appear on any possible locations. For images with multiple color channels, each channel are masked in the same manner.

**Relations to the spatial masking in MAE**. MAE aims to encode an image patch into a feature vector, which is a downsampling process and does not require high precision along spatial dimensions (*i.e.*, pixels within a patch are considered as the same). GAN, on the contrary, aims for decoding a latent code into a high-resolution image, which is an upsampling process and requires high fidelity along

spatial dimensions (*i.e.*, pixels within a patch should not be completely the same). We therefore introduce an additional "mask shift" to allow the masked patches to appear at any image locations (instead of fixed grid locations only as in the spatial masking in MAE [21, 2]), which helps generate images with high-fidelity along spatial dimensions.

**Balanced spectral masking** masks random spectral bands of an image in the decomposed spectral space, aiming to encourage the discriminator to learn from all spectral bands instead of focusing on easy bands (*e.g.*, low-frequency bands capturing color and brightness) only. Specifically, *balanced spectral masking* decomposes an image into multiple bands in spectral space and masks a portion of spectral bands with self-adaptive probabilities, *i.e.,* a spectral band is masked with a higher (or lower) probability if it contains more (or less) contents. The balanced masking is crucial to the spectral masking as image contents are usually unevenly distributed along the spectral dimension. Specifically, the contents of natural images are distributed very densely around low-frequency bands but very sparsely in high-frequency bands. Therefore, uniform masking along the spectral dimension is imbalanced and does not work well for image generation.

Given an image $x \in \mathbb{R}^{H \times W}$, we first employ Fourier transformation [53, 23, 24, 56] to decompose it into multi-band representation $x_{\text{mul}} \in \mathbb{R}^{H \times W \times C}$, where $C$ denotes the number of spectral bands. The image contents in each band $I_x \in \mathbb{R}^C$ can then be computed by summing all contents in each band, *i.e.*, $I_x = \sum_H \sum_W x_{\text{mul}}^{(H,W,C)}$, and followed by an normalization operation: $I'_x = I_x^{(C)} / \sum_C I_x^{(C)}$. With the calculated $I'_x$, The balanced spectral mask $m_{\text{spectral}} \in \{0, 1\}^C$ can be generated by masking $c$-th band with probability $I_x'^{(c)}$. Finally, the balanced spectral masking function can be defined by: $\mathbb{M}_{\text{spectral}}(x) = \sum_C (x_{\text{mul}}^{(H,W,C)} \times m_{\text{spectral}}^{(C)})$.

**Loss functions.** We apply the two image masking strategies to both discriminator and generator training, where the overall training losses can be formulated as follows:

$$\mathcal{L}_d = \mathop{\mathbb{E}}_{x \sim p_{\text{data}}} [f_d(-D(\mathbb{M}(x)))] + \mathop{\mathbb{E}}_{z \sim p_z} [f_d(D(\mathbb{M}(G(z))))], \tag{3}$$

$$\mathcal{L}_g = \mathop{\mathbb{E}}_{z \sim p_z} [f_g(-D(\mathbb{M}(G(z))))], \tag{4}$$

where $\mathbb{M}(\cdot) = \mathbb{M}_{\text{spectral}}(\mathbb{M}_{\text{spatial}}(\cdot))$ is the combination of *shifted spatial masking* and *balanced spectral masking*.

Note the two masking strategies work in different manners in the discriminator and generator training. Specifically, by applying image masking on input images in discriminator training, we could create a challenging discriminator task that requires holistic learning beyond certain easy-to-discriminate information. Differently, in generator training, the image masking operation works on output side by sparsifying the training signals back-propagated from the discriminator. One major consideration is to slow down the generator training so that it has similar learning pace as the discriminator which further stabilizes the overall adversarial learning between the discriminator and the generator.

### 3.3 Theoretical Insights

The proposed Masked Generative Adversarial Network (MaskedGAN) is inherently connected with the theory of stochastic approximation.

**Proposition 1** *The MaskedGAN can be modeled as an instance of the Two Time-Scale Update Rule.*

**Proposition 2** *The MaskedGAN converges to a Local Nash Equilibrium under certain conditions.*

The proofs of **Propositions 1** and **2** are provided in the Appendix.

## 4 Experiments

We evaluate MaskedGAN over different network architectures and public datasets. Sections 4.1 presents experiments with BigGAN over datasets CIFAR-10 [34], CIFAR-100 [34] and ImageNet [15]. Section 4.2 reports experimental results with StyleGAN-v2 over datasets 100-shot [58], AFHQ [11] and FFHQ [32]. Section 4.3 presents experiments with two transformer-based GANs (TransGAN and

Table 1: **Conditional image generation with BigGAN on CIFAR-10 and CIFAR-100.** MaskedGAN (*i.e.*, Masked-BigGAN) achieves superior performance especially with limited training data. We calculate FID ($\downarrow$) scores with 10K generated samples and the validation set, as in [58].

| Method | CIFAR-10 | | | CIFAR-100 | | |
|---|---|---|---|---|---|---|
| | 100% Data | 20% Data | 10% Data | 100% Data | 20% Data | 10% Data |
| Non-saturated GAN [19] | 9.83 ± 0.06 | 18.59 ± 0.15 | 41.99 ± 0.18 | 13.87 ± 0.08 | 32.64 ± 0.19 | 70.5 ± 0.38 |
| LS-GAN [36] | 9.07 ± 0.01 | 21.60 ± 0.11 | 41.68 ± 0.18 | 12.43 ± 0.11 | 27.09 ± 0.09 | 54.69 ± 0.12 |
| RAHinge GAN [29] | 11.31 ± 0.04 | 23.90 ± 0.22 | 48.13 ± 0.33 | 14.61 ± 0.21 | 28.79 ± 0.17 | 52.72 ± 0.18 |
| StyleGAN-v2 [33] | 11.07 ± 0.03 | 23.08 ± 0.11 | 36.02 ± 0.15 | 16.54 ± 0.04 | 32.30 ± 0.11 | 45.87 ± 0.15 |
| BigGAN [4] (baseline) | 9.74 ± 0.06 | 21.86 ± 0.29 | 48.08 ± 0.10 | 13.60 ± 0.07 | 32.99 ± 0.24 | 66.71 ± 0.01 |
| LeCam-GAN [46] | 8.31 ± 0.05 | 15.27 ± 0.10 | 35.23 ± 0.14 | 11.88 ± 0.12 | 25.51 ± 0.19 | 49.63 ± 0.16 |
| GenCo [13] | 8.83 ± 0.04 | 16.57 ± 0.08 | 28.08 ± 0.11 | 11.90 ± 0.02 | 26.15 ± 0.08 | 40.98 ± 0.09 |
| ADA [31] | 8.99 ± 0.03 | 19.87 ± 0.09 | 30.58 ± 0.11 | 12.22 ± 0.02 | 22.65 ± 0.10 | 27.08 ± 0.15 |
| DA [58] | 8.75 ± 0.03 | 14.53 ± 0.10 | 23.34 ± 0.09 | 11.99 ± 0.02 | 22.55 ± 0.06 | 35.39 ± 0.08 |
| **MaskedGAN** | 8.41 ± 0.03 | 12.51 ± 0.09 | 15.89 ± 0.12 | 11.65 ± 0.03 | 18.33 ± 0.09 | 24.02 ± 0.12 |

GANformer) over datasets CIFAR-10, CIFAR-100 and Cityscapes [12]. In addition, we present ablation studies in Section 4.1 and discuss different features of the proposed MaskedGAN in Section 4.4 The implementation and dataset details are provided in appendix.

## 4.1 Conditional image generation with BigGAN on CIFAR-10, CIFAR-100 and ImageNet

Table 2: **Ablation studies of MaskedGAN** with BigGAN (baseline) on 10% CIFAR-10 data.

| Method | Shifted Spatial Masking | | Balanced Spectral Masking | | FID ($\downarrow$) |
|---|---|---|---|---|---|
| | Spatial Masking | Random Shift | Spectral Masking | Self-adaptive Probability | |
| BigGAN [4] | | | | | 48.08 |
| | ✓ | | | | 28.35 |
| | ✓ | ✓ | | | 22.56 |
| | | | ✓ | | 35.20 |
| | | | ✓ | ✓ | 28.78 |
| **MaskedGAN** | ✓ | ✓ | ✓ | ✓ | 15.89 |

Table 3: **Conditional image generation with BigGAN on ImageNet.** MaskedGAN (*i.e.*, Masked-BigGAN) improves the image generation consistently across different setups of the large-scale ImageNet. We evaluate the models with 10K generated samples and the whole training set.

| Method | 10% training data | | 5% training data | | 2.5% training data | |
|---|---|---|---|---|---|---|
| | IS↑ | FID↓ | IS↑ | FID↓ | IS↑ | FID↓ |
| BigGAN [4] (baseline) | 10.94 ± 0.35 | 38.30 ± 0.25 | 6.13 ± 0.09 | 91.16 ± 0.43 | 3.92 ± 0.07 | 133.80 ± 0.76 |
| ADA [31] | 12.67 ±0.31 | 31.89 ±0.17 | 9.44 ±0.25 | 43.21 ± 0.37 | 8.54 ± 0.26 | 56.83 ± 0.48 |
| DA [58] | 12.76 ± 0.34 | 32.82 ± 0.18 | 9.63 ± 0.21 | 56.75 ± 0.35 | 8.17 ± 0.28 | 63.49 ± 0.51 |
| **MaskedGAN** | 13.34 ± 0.24 | 26.51 ± 0.12 | 12.85 ± 0.40 | 35.70 ± 0.31 | 12.68 ± 0.27 | 38.62 ± 0.37 |

**CIFAR-10 and CIFAR-100.** Table 1 reports class-conditional image generation results with baseline BigGAN [4] on CIFAR-10 and CIFAR-100. All models are trained with 100%, 20% or 10% training data (*i.e.*, 50K, 10K or 5K images), and evaluated over the validation set (10K images). It shows that MaskedGAN (*i.e.*, Masked-BigGAN) achieves superior performance over CIFAR-10 and CIFAR-100 as compared with state-of-the-art methods. Results are averaged with three runs.

**Ablation studies over CIFAR-10.** We perform ablation studies with the widely adopted BigGAN [4] over 10% CIFAR-10 data as shown in Table 2. As the core of the proposed MaskedGAN, we examine how our designed *shifted spatial masking* and *balanced spectral masking* contribute to the overall performance of data-limited image generation. As shown in Table 2, the *baseline* (BigGAN) in the first row does not perform well with limited training data. Including the basic *Spatial Masking* in

Table 4: **Unconditional image generation with StyleGAN-v2 on 100-shot.** With only 100 training images, MaskedGAN (*i.e.*, Masked-StyleGAN-v2) outperforms the baseline StyleGAN-v2 significantly in FID (↓). It also outperforms the state-of-the-art consistently, with and without pre-training.

| Methods | Pre-training w/ 70K images | 100-shot Obama | Grumpy Cat | Panda |
|---|---|---|---|---|
| Scale/shift [41] | Yes | 50.72 | 34.20 | 21.38 |
| MineGAN [49] | Yes | 50.63 | 34.54 | 14.84 |
| TransferGAN [50] | Yes | 48.73 | 34.06 | 23.20 |
| TransferGAN + DA [58] | Yes | 39.85 | 29.77 | 17.12 |
| FreezeD [40] | Yes | 41.87 | 31.22 | 17.95 |
| StyleGAN-v2 [33] (baseline) | No | 80.20 | 48.90 | 34.27 |
| ADA [31] | No | 45.69 | 26.62 | 12.90 |
| LeCam-GAN [46] | No | 38.58 | 41.38 | 19.88 |
| GenCo [13] | No | 36.35 | 33.57 | 15.50 |
| AdvAug [9] | No | 52.86 | 31.02 | 14.75 |
| DA [58] | No | 46.87 | 27.08 | 12.06 |
| APA [27] | No | 43.75 | 28.49 | 12.34 |
| InsGen [52] | No | 45.85 | 27.48 | 12.13 |
| **MaskedGAN** | No | 33.78 ± 0.27 | 20.06 ± 0.13 | 8.93 ± 0.06 |

Table 5: **Unconditional image generation with StyleGAN-v2 on AFHQ and FFHQ.** MaskedGAN (*i.e.*, Masked-StyleGAN-v2) works effectively with limited training data over face generation tasks. We calculate FID (↓) scores with 10K generated samples and the whole training set.

| Method | AFHQ-Cat | | | AFHQ-Dog | | |
|---|---|---|---|---|---|---|
| | 20% Data | 10% Data | 5% Data | 20% Data | 10% Data | 5% Data |
| StyleGAN-v2 [33] (baseline) | 12.51 ± 0.09 | 17.67 ± 0.11 | 26.27 ± 0.21 | 43.74 ± 0.29 | 56.84 ± 0.36 | 95.32 ± 0.51 |
| **MaskedGAN** | 12.03 ± 0.10 | 15.27 ± 0.09 | 19.82 ± 0.13 | 42.25 ± 0.22 | 46.28 ± 0.25 | 46.68 ± 0.27 |

| Method | AFHQ-Wild | | | FFHQ | | |
|---|---|---|---|---|---|---|
| | 20% Data | 10% Data | 5% Data | 5% Data | 2.5% Data | 1.25% Data |
| StyleGAN-v2 [33] (baseline) | 8.61 ± 0.03 | 14.82 ± 0.11 | 33.85 ± 0.21 | 30.06 ± 0.25 | 47.90 ± 0.31 | 127.82 ± 0.58 |
| **MaskedGAN** | 7.76 ± 0.02 | 10.39 ± 0.09 | 15.70 ± 0.11 | 28.53 ± 0.14 | 35.04 ± 0.21 | 44.36 ± 0.24 |

GAN training improves the baseline clearly as shown in the 2nd row, while further involving *Random Shift* on top of *Spatial Masking* brings clear improvement in FID as shown in the 3rd row. On the other hand, the basic *Spectral Masking* (masks spectral bands with the same probability) does not work well as shown in the 4th row as image contents are usually unevenly distributed along the spectral dimension. However, *Balanced spectral masking* which includes the proposed *Self-adaptive Probabilities* as shown in the 5th row, improves image generation clearly in FID (19.30 above the baseline), showing that the balanced masking strategy is crucial for effective spectral masking. In addition, the two image masking strategies work in orthogonal dimensions (*i.e.*, spatial and spectral dimensions), which complement each other in data-limited image generation. We can observe clearly that combining these two masking strategies (*MaskedGAN* in the last row) performs the best.

**ImageNet.** Table 3 reports class-conditional generation results with BigGAN [4] over ImageNet at 64×64 resolution. The models are trained with different amounts of training data including ~10% (100K images), ~5% (50K images) and ~2.5% (25K images), and evaluated over the whole training set (~1.2 million images). Experiments show that MaskedGAN works effectively with different data setups over the very diverse and large-scale ImageNet. Results are averaged with three runs.

### 4.2 Unconditional image generation with StyleGAN-v2 on 100-shot, AFHQ and FFHQ

**100-shot.** Table 4 reports unconditional image generation results with StyleGAN-v2 [4] on 100-shot dataset. All models are trained and evaluated over 100 images of Obama, Grumpy Cat and Panda,

Table 6: **Unconditional image generation with TransGAN on CIFAR-10 and CIFAR-100.** MaskedGAN works effectively with transformer-based GAN (TransGAN) as well. We calculate FID (↓) scores with 10K generated samples and the validation set.

| Method | CIFAR-10 | | | CIFAR-100 | | |
|---|---|---|---|---|---|---|
| | 100% Data | 20% Data | 10% Data | 100% Data | 20% Data | 10% Data |
| TransGAN [28] (baseline) | $22.53 \pm 0.18$ | $53.58 \pm 0.31$ | $68.59 \pm 0.40$ | $25.73 \pm 0.22$ | $58.61 \pm 0.37$ | $73.95 \pm 0.42$ |
| **MaskedGAN** | $8.71 \pm 0.09$ | $18.29 \pm 0.13$ | $29.23 \pm 0.21$ | $16.88 \pm 0.15$ | $26.17 \pm 0.23$ | $32.28 \pm 0.25$ |

Table 7: **Unconditional image generation with GANformer on Cityscapes.** MaskedGAN brings significant gains over the baseline GANformer consistently. We calculate the FID (↓) scores with 10K generated samples and the whole training set.

| Method | 5% Training Data | | 2.5% training data | | 1.25% training data | |
|---|---|---|---|---|---|---|
| | IS↑ | FID↓ | IS↑ | FID↓ | IS↑ | FID↓ |
| GANformer [26] (baseline) | $1.61 \pm 0.02$ | $34.87 \pm 0.25$ | $1.56 \pm 0.02$ | $62.23 \pm 0.42$ | $1.44 \pm 0.01$ | $108.73 \pm 0.58$ |
| **MaskedGAN** | $1.64 \pm 0.02$ | $29.84 \pm 0.21$ | $1.60 \pm 0.02$ | $55.74 \pm 0.32$ | $1.56 \pm 0.02$ | $75.75 \pm 0.45$ |

respectively. Experimental results show that the proposed MaskedGAN works effectively with only 100 images, *i.e.*, it brings significant gains over StyleGAN-v2 (baseline) and outperforms state-of-the-art methods by clear margins. Besides, MaskedGAN surpasses all transfer learning methods but does not require pre-training over large-scale datasets. Results are averaged with three runs.

**AFHQ and FFHQ.** Table 5 presents unconditional image generation results with StyleGAN-v2 [4] on AFHQ and FFHQ. For AFHQ, the models are trained with different amounts of training data including ~20% (1K images), ~10% (500 images) and ~5% (250 images), and evaluated over the whole training set ( 5K images). For FFHQ, the models are trained with ~10% (7K images), ~5% (3.5K images) and ~2.5% (1.75K images) data, and evaluated over the whole training set (70K images). It can be observed that MaskedGAN works effectively with limited training data over animal and human face generation tasks. Results are averaged with three runs.

## 4.3 Unconditional image generation with TransGAN and GANformer

We evaluate MaskedGAN with recent transformer-based architectures including TransGAN [28] and GANformer [26]. Specifically, TransGAN is the first work that uses purely transformers for image generation. GANformer employs a bipartite structure for computing soft attention, which circumvents the computation constraints in standard transformers and leads to improved image generation.

**CIFAR-10 and CIFAR-100 with TransGAN.** Tables 6 reports unconditional image generation with TransGAN on CIFAR-10 and CIFAR-100. The models are trained with 100%, 20% or 10% training data (*i.e.*, 50K, 10K or 5K images), and evaluated over the whole training set (50K images). We can observe that our image masking strategies work effectively with transformer-based GAN as well. Results are acquired with three runs.

**Cityscapes with GANformer.** Tables 7 reports unconditional image generation with GANformer on Cityscapes. The models are trained with ~5%, ~2.5% or ~1.25% training data (*i.e.*, 1.25K, 0.63K or 0.31K images ), and evaluated over the whole training set (*i.e.*, ~25K images ). Experiments show that our image masking strategies bring significant gains over the baseline GANformer consistently over scene generation task. Results are acquired with three runs.

## 4.4 Discussion

**Generalization across different GAN architectures:** We examine the generalization of the proposed MaskedGAN by evaluating it with four representative GAN architectures, including two CNN-based (*i.e.*, BigGAN and StyleGAN-v2) and two Transformer-based (*i.e.*, TransGAN and GANformer). Experimental results in Tables 1- 7 show that our MaskedGAN brings clear generation improvements consistently over different GAN architectures.

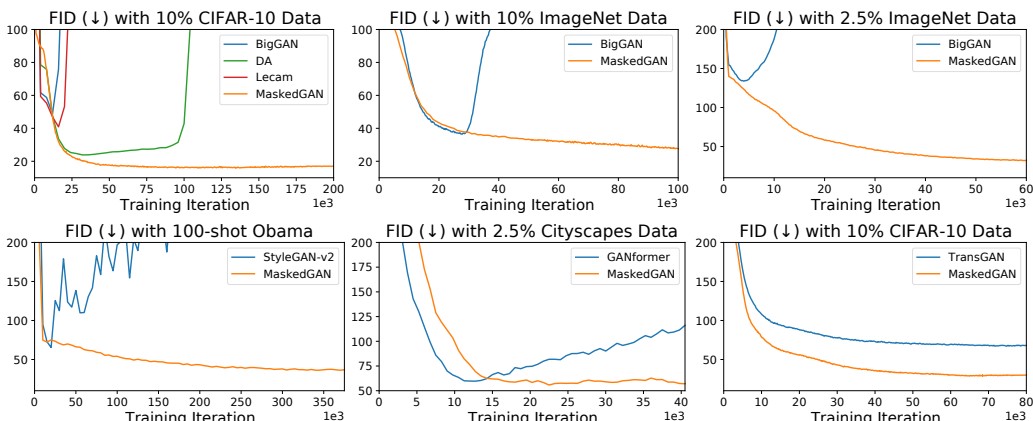

Figure 3: **Convergence of GAN training with limited training data:** MaskedGAN converges well consistently across different datasets (*e.g.*, CIFAR-10, ImageNet, 100-shot and Cityscapes), different amounts of training data (*e.g.*, 10% and 2.5% of ImageNet), and different generation architectures (*e.g.*, BigGAN, StyleGAN-v2, GANformer and TransGAN). Most existing GANs instead experience clear training collapses with limited training data. The recent TransGAN can converge but it trains severely degraded generation model (~68 in FID).

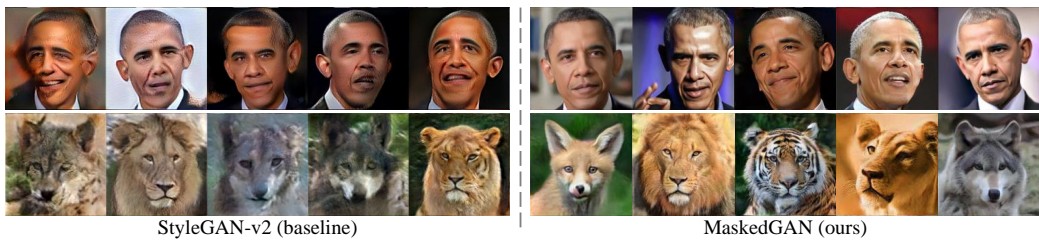

StyleGAN-v2 (baseline)                    MaskedGAN (ours)

Figure 4: **Qualitative illustration and comparison** over 100-shot Obama (top) and AFHQ-Wild with 5% data (bottom): With limited training data, introducing our image masking into GAN training helps generate more realistic and high-fidelity images, especially in terms of image shapes and structures.

**Generalization across different generation tasks:** We study the generalization of the proposed MaskedGAN from the perspective of image generation tasks. Specifically, we perform extensive evaluations over object generation tasks with CIFAR-10, CIFAR-100, and ImageNet, face generation tasks with 100-shot AFHQ, and FFHQ, and scene generation tasks with Cityscapes. Experimental results in Tables 1- 7 show that the proposed MaskedGAN achieves superior generation consistently over different generation tasks.

**Generalization over different numbers of training samples:** We study the generalization of the proposed MaskedGAN from the perspective of the number of training samples. Specifically, we benchmark MaskedGAN on CIFAR-10 and CIFAR-100 with 100%, 20% and 10% data, ImageNet with 10%, 5% and 2.5% data, AFHQ with 20%, 10% and 5% data , FFHQ and Cityscapes with 5%, 2.5% and 1.25% data and 100-shot dataset with only 100 images. Experiments in Tables 1- 7 show that MaskedGAN achieves superior performance consistently with different amounts of training data.

**Convergence comparison across different network architectures and datasets:** We examine the convergence of the proposed MaskedGAN by benchmarking it over various network architectures (*e.g.*, CNNs-based including BigGAN and StyleGAN-v2 and Transformers-based including TransGAN and GANformer) and datasets (*e.g.*, CIFAR-10, ImageNet, 100-shot and Cityscapes). Fig. 3 provides the line charts of FID versus *training iteration*. It shows clearly that most state-of-the-art GANs [33, 4] experience generation failures (or called training collapses) with severely degraded generation performance when only limited training data is available. In addition, prior studies on data-limited generation (*e.g.*, data augmentation method [58] and model regularization method [46]) could improve the performance in some degree but still suffer from generation failures and training collapses. As a comparison, the proposed MaskedGAN converges well consistently across different amounts of

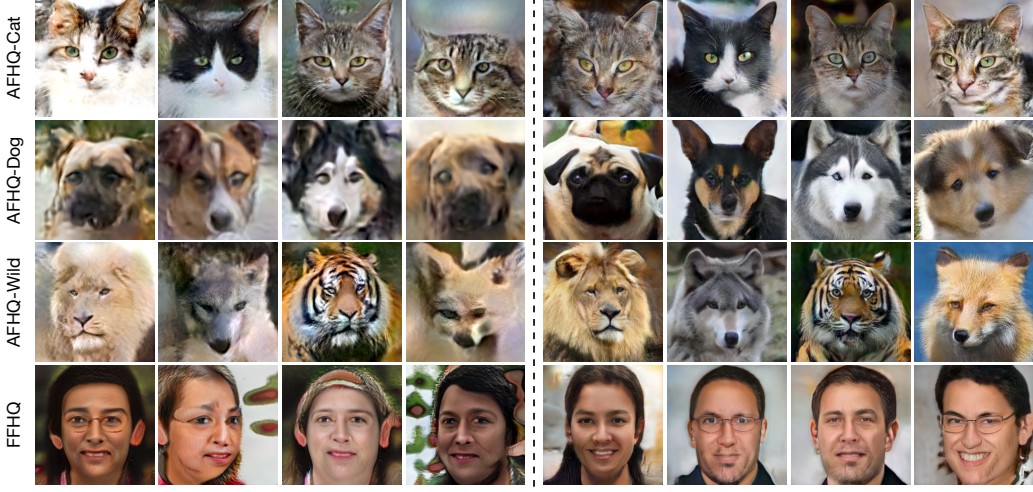

Figure 5: **Qualitative illustration and comparison** over AFHQ-Cat, AFHQ-Dog and AFHQ-Wild with 5% training data and FFHQ with 1.25% data: With limited training data, introducing the proposed image masking strategies into GAN training helps generate more realistic and high-fidelity images, especially in terms of image shapes and structures.

training data, network architectures and datasets. The great convergence of MaskedGAN is largely attributed to two factors: its image masking designs suppress trivial solutions and training failures directly; it keeps similar learning paces for discriminator and generator which ensures that network converge to a Local Nash Equilibrium under certain conditions [22].

We provide qualitative illustrations in Fig. 4 and 5, which show that MaskedGAN outperforms the baseline and state-of-the-art models clearly. Specifically, with limited training data, introducing the proposed image masking strategies into GAN training helps generate more realistic and high-fidelity images, especially in terms of image shapes and structures. Due to the space limit, we provide additional discussions and qualitative illustrations (including comparisons with the state-of-the-arts) in appendix.

## 5   Conclusion

This paper presents MaskedGAN, a robust image generation network that can learn effectively with limited training data. MaskedGAN introduces two image masking strategies that work along orthogonal dimensions of training images, namely, *shifted spatial masking* and *balanced spectral masking*. The two masking strategies complement each other which together encourage more challenging holistic learning from limited training data, ultimately suppressing trivial solutions and failures in GAN training. Extensive experiments show that MaskedGAN achieves superior performance consistently across different network architectures (*e.g.*, CNNs including BigGAN and StyleGAN-v2 and Transformers including TransGAN and GANformer) and datasets (*e.g.*, CIFAR-10, CIFAR-100, ImageNet, 100-shot, AFHQ, FFHQ and Cityscapes). Moving forward, we will explore data masking training in other generation tasks such multi-modality generation.

## Acknowledgement

This study is supported under the RIE2020 Industry Alignment Fund – Industry Collaboration Projects (IAF-ICP) Funding Initiative, as well as cash and in-kind contribution from Singapore Telecommunications Limited (Singtel), through Singtel Cognitive and Artificial Intelligence Lab for Enterprises (SCALE@NTU). This research was also carried out on the High Performance Computing resources at Inception Institute of Artificial Intelligence, Abu Dhabi.

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
