# Masked Generative Adversarial Networks are Data-Efficient Generation Learners
# Supplemental Materials

**Jiaxing Huang**[1]*, **Kaiwen Cui**[1]*, **Dayan Guan**[1], **Aoran Xiao**[1], **Fangneng Zhan**[2],
**Shijian Lu**[1]†, **Shengcai Liao**[3], **Eric Xing**[45]

[1] School of Computer Science and Engineering, Nanyang Technological University, Singapore
[2] Max Planck Institute for Informatics, Germany
[3] Inception Institute of Artificial Intelligence (IIAI), Abu Dhabi, UAE
[4] School of Computer Science, Carnegie Mellon University, USA
[5] Mohamed bin Zayed University of Artificial Intelligence, Abu Dhabi, UAE
{Jiaxing.Huang, Kaiwen.Cui, Dayan.Guan, Aoran.Xiao, Shijian.Lu}@ntu.edu.sg
fzhan@mpi-inf.mpg.de, shengcai.liao@inceptioniai.org
epxing@cs.cmu.edu, Eric.Xing@mbzuai.ac.ae

## A    GAN training failures with limited data

**GAN training failures with limited data**. Prior studies [18, 12] show that GAN often experiences generation failures with severely degraded generation performance when only limited training data is available. Specifically, with limited training data, the discriminator tends to discriminate via meaningless shortcuts by merely focusing on easy-to-discriminate image locations and spectra instead of holistic understanding of images. This can be viewed clearly in Fig. 1, where the Gini Coefficient [4] of discriminator's spatial attentions increases quickly along the training iteration (when only limited training data is available). Note that the Gini coefficient [4] is negatively correlated with equality, *i.e.*, the discriminator will pay more unevenly distributed attention to each spatial location while the Gini coefficient increases from '0' to '1'. For image generation with GAN, the large Gini coefficient (of discriminator's spatial attentions) thus means that the discriminator starts to focus on certain spatial locations (easy to discriminate) while ignoring other spatial locations (hard to discriminate), ultimately leading to an over-confident discriminator and training collapse. In another word, the Gini coefficient [4] of '0' expresses perfect equality where all values are the same (*i.e.*, where the discriminator pays the same attention to every spatial location) while '1' expresses maximal inequality among values (*i.e.*, the discriminator focuses on only one location while all others are ignored).

## B    Theoretical insights of MaskedGAN

The Two Time-Scale Update Rule (TTUR) [8] was proposed to prove the convergence of GAN training: based on the theory of stochastic approximation, [8] proved that the GANs trained with TTUR converge to a stationary Local Nash Equilibrium under certain conditions. In this section, we show that the MaskedGAN can be modeled as an instance of the Two Time-Scale Update Rule.

Specifically, TTUR assigns an individual learning rate for discriminator and generator such that GAN can converge to a local Nash equilibrium when the learning rate of generator is far smaller than that of discriminator (*i.e.*, the generator changes slowly enough) [8]. In MaskedGAN, we slow

---

*indicates equal contribution.

†corresponding author.

36th Conference on Neural Information Processing Systems (NeurIPS 2022).

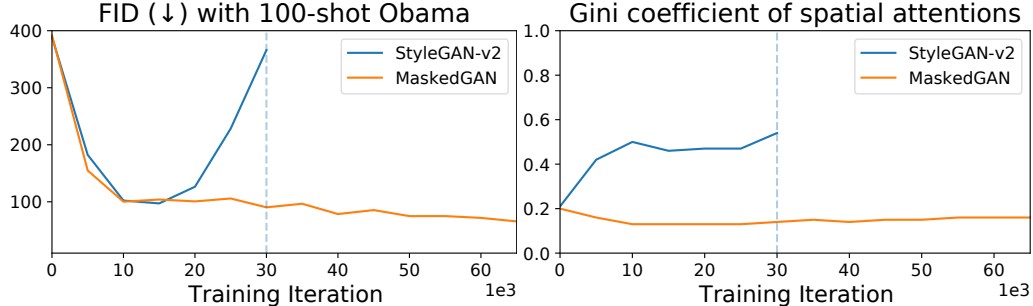

Figure 1: **GAN training failures with limited data:** In state-of-the-art GANs such as StyleGAN-v2, the discriminator tends to discriminate via certain less meaningful shortcuts by merely focusing on some easy-to-discriminate image information without holistic understanding of images. This can be viewed clearly in the right graph where the Gini Coefficient [4] of discriminator's spatial attentions increases quickly along the training iteration, which shows that the discriminator starts to focus on certain spatial locations (*e.g.*, easy to discriminate ones) while ignoring other spatial locations (*e.g.*, hard to discriminate ones), ultimately leading to an over-confident discriminator and training collapse. Specifically, it shows a randomly initialized model produces very even spatial attentions, i.e., a Gini coefficient/index of '0.2'. With limited training data, the discriminator in StyleGAN-v2 tends to focus on certain spatial locations, where the Gini coefficient of spatial attentions increases to '~0.5' which indicates severe attention inequality. As a comparison, our MaskedGAN pays very even attention on every spatial location along the training process (*i.e.*, around 0.15), resulting in more stable training process and better performance. The major reason is that MaskedGAN encourages holistic learning of images, such as using spatial masking to enforce discriminator to learn over all image locations instead of merely focusing on easy-to-discriminate locations. Note the FID scores lags behind Gini coefficients in training iteration as there is a delay for the side effects caused by the discriminator to take effects on the generator.

down the generator learning by masking out (or blocking) partial training signals. Specifically, in the generator training, image masking works by sparsifying the training signals backpropagated from the discriminator.

Since our MaskedGAN introduces a general idea of randomly masking certain image information in GAN training, to facilitate this theoretical analysis, we consider masking image features [16, 1, 7]. We consider the standard min-max GAN loss function $f_g(x) = (1 + x)$ and the simple spatial masking function $\mathbb{M}(x) = m_{\text{spatial}} \times x$ ($m_{\text{spatial}} \in \{0, 1\}^{H \times W}$). We consider a discriminator $D(\cdot) = C(F(\cdot))$, where $C(\cdot)$ is a linear classifier and $F(\cdot)$ is a feature encoder. Let $p_z(z)$ be the distribution of latent vector $z$, and $G(\cdot)$ be the generator. Given a fixed discriminator, we have the conventional generator training loss $\mathcal{L}_g$ (without masking) and the MaskedGAN generator training loss $\mathcal{L}_g^{\text{masked}}$ (with masking) defined as follows, respectively:

$$\begin{aligned}
\mathcal{L}_g &= \mathop{\mathbb{E}}_{z \sim p_z(z)} [f_g(-D(G(z)))] \\
&= \mathop{\mathbb{E}}_{z \sim p_z(z)} [(1 - D(G(z)))] \\
&= \mathop{\mathbb{E}}_{z \sim p_z(z)} [(1 - C(F(G(z))))], \quad (1) \\
\mathcal{L}_g^{\text{masked}} &= \mathop{\mathbb{E}}_{z \sim p_z(z)} [(1 - C(\mathbb{M}(F(G(z)))))], \quad (2)
\end{aligned}$$

Then, let us consider modelling MaskedGAN as an instance of the Two Time-Scale Update Rule.

**Proposition 1** *The MaskedGAN can be modeled as an instance of the Two Time-Scale Update Rule.*

*Proof:* Given any discriminator $D$, the two generator training losses $\mathcal{L}_g$ and $\mathcal{L}_g^{\text{masked}}$ (in Eq.1 and Eq.2) can be re-written as follows, respectively:

$$\mathcal{L}_g = \int_z p_z(z)(1 - C(F(G(z)))) \, dz, \tag{3}$$

$$\mathcal{L}_g^{\text{masked}} = \int_z p_z(z)(1 - C(\mathbb{M}(F(G(z))))) \, dz, \tag{4}$$

Let $f_z = F(G(z))$, Eqs.3-4 can be simplified as follows:

$$\mathcal{L}_g = \int_z p_z(z)(1 - C(f_z)) \, dz, \tag{5}$$

$$\mathcal{L}_g^{\text{masked}} = \int_z p_z(z)(1 - C(\mathbb{M}(f_z))) \, dz, \tag{6}$$

We consider the linear classifier $C(\cdot)$ in [2], which consists of a global spatial pooling layer followed by a linear layer, *i.e.*, $C(x) = a \sum_{\text{spatial}} x + b$. Then, we have:

$$\mathcal{L}_g = \int_z p_z(z)(1 - a \sum_{\text{spatial}} f_z + b) \, dz, \tag{7}$$

$$\mathcal{L}_g^{\text{masked}} = \int_z p_z(z)(1 - a \sum_{\text{spatial}} \mathbb{M}(f_z) + b) \, dz, \tag{8}$$

where $a$ is the linear weight, $b$ is the linear bias and $\sum_{\text{spatial}}$ is the global spatial pooling operation.

As function $\mathbb{M}(x) = m_{\text{spatial}} \times x$ masks images with the randomly generated mask $m_{\text{spatial}}$, let $M = \{m_{\text{spatial}}^i\}_{i=1}^I$ denote all possible masks, we can re-write Eq.8 as follow:

$$\mathcal{L}_g^{\text{masked}} = \sum_{i \in I} p_i(i) \int_z p_z(z)(1 - a \sum_{\text{spatial}} m_{\text{spatial}}^i f_z + b) \, dz \tag{9}$$

$$= \int_z p_z(z)(\sum_{i \in I} p_i(i)(1 - a \sum_{\text{spatial}} m_{\text{spatial}}^i f_z + b)) \, dz \tag{10}$$

$$= \int_z p_z(z)(1 - \sum_{i \in I} p_i(i) a \sum_{\text{spatial}} m_{\text{spatial}}^i f_z + b) \, dz \tag{11}$$

$$= \int_z p_z(z)(1 - a \sum_{\text{spatial}} \sum_{i \in I} p_i(i) m_{\text{spatial}}^i f_z + b) \, dz, \tag{12}$$

where $I$ denotes the number of all possible permutation of binary mask $m_{\text{spatial}}$, $m_{\text{spatial}}^i$ denotes the $i$-th mask and $p_i(i)$ stands for the probability of $m_{\text{spatial}}^i$. In Eq.9, we ignore the element-wise multiplication symbol '$\times$' for simplicity.

Since the spatial mask $m_{\text{spatial}}$ (in masking function $\mathbb{M}(x) = m_{\text{spatial}} \times x$) is generated uniformly randomly with a fixed overall masking percentage $p_g^{\text{mask}}$, we have:

$$p_i(i) = \frac{1}{I}, \ \forall i \in I, \tag{13}$$

$$\sum_{i \in I} p_i(i) m_{\text{spatial}}^i f_z = \sum_{i \in I} \frac{1}{I} m_{\text{spatial}}^i f_z \tag{14}$$

$$= \frac{1}{I} \sum_{i \in I} m_{\text{spatial}}^i f_z \tag{15}$$

$$= (\frac{1}{I} \sum_{i \in I} m_{\text{spatial}}^i) f_z \tag{16}$$

$$= (1 - p_g^{\text{mask}}) f_z, \tag{17}$$

where $m_{\text{spatial}}^i$ is a binary spatial mask that is randomly generated with uniform probabilities (for all locations) and a fixed overall masking percentage $p_g^{\text{mask}}$.

Substituting Eq.17 into Eq.12, we can re-write the MaskedGAN generator training loss $\mathcal{L}_g^{\text{masked}}$ as follow:

$$\mathcal{L}_g^{\text{masked}} = \int_z p_z(z)(1 - a \sum_{\text{spatial}} (1 - p_g^{\text{mask}}) f_z + b) \, dz, \tag{18}$$

where $p_g^{\text{mask}}$ is the overall masking ratio (or called overall masking percentage).

By using the Sum Rule of Integration, we can simplify Eq.7 and Eq.18 as follows:

$$\mathcal{L}_g = \int_z p_z(z)(1 - a \sum_{\text{spatial}} f_z + b) \, dz \tag{19}$$

$$= \int_z p_z(z)(1 + b) \, dz - \int_z p_z(z) a \sum_{\text{spatial}} f_z \, dz, \tag{20}$$

$$\mathcal{L}_g^{\text{masked}} = \int_z p_z(z)(1 - a \sum_{\text{spatial}} (1 - p_g^{\text{mask}}) f_z + b) \, dz \tag{21}$$

$$= \int_z p_z(z)(1 + b) \, dz - \int_z p_z(z) a \sum_{\text{spatial}} (1 - p_g^{\text{mask}}) f_z \, dz \tag{22}$$

$$= \int_z p_z(z)(1 + b) \, dz - (1 - p_g^{\text{mask}}) \int_z p_z(z) a \sum_{\text{spatial}} f_z \, dz, \tag{23}$$

Based on Eq.20 and Eq.23, we can derive the gradients of the generator loss functions $\mathcal{L}_g$ and $\mathcal{L}_g^{\text{masked}}$ as follows, respectively:

$$\nabla_{\theta_g} \mathcal{L}_g = \nabla_{\theta_g} [\int_z p_z(z)(1 + b) \, dz - \int_z p_z(z) a \sum_{\text{spatial}} f_z \, dz] \tag{24}$$

$$= -\nabla_{\theta_g} \int_z p_z(z) a \sum_{\text{spatial}} f_z \, dz, \tag{25}$$

$$\nabla_{\theta_g} \mathcal{L}_g^{\text{masked}} = \nabla_{\theta_g} [\int_z p_z(z)(1 + b) \, dz - (1 - p_g^{\text{mask}}) \int_z p_z(z) a \sum_{\text{spatial}} f_z \, dz] \tag{26}$$

$$= -\nabla_{\theta_g} (1 - p_g^{\text{mask}}) \int_z p_z(z) a \sum_{\text{spatial}} f_z \, dz \tag{27}$$

$$= -(1 - p_g^{\text{mask}}) \nabla_{\theta_g} \int_z p_z(z) a \sum_{\text{spatial}} f_z \, dz, \tag{28}$$

where $\theta_g$ stands for the parameters of generator $G$.

By comparing Eq.25 and Eq.28, we have:

$$\nabla_{\theta_g} \mathcal{L}_g^{\text{masked}} = (1 - p_g^{\text{mask}}) \nabla_{\theta_g} \mathcal{L}_g \tag{29}$$

Since $p_g^{\text{mask}} \in [0, 1]$, Eq.25 and Eq.28 indicate that MaskedGAN can slows down the generator learning by scaling down the gradient of the generator loss function. For example, **1)** if the masking ratio is set as '0' (*i.e.,* $p_g^{\text{mask}} = 0$ and $(1 - p_g^{\text{mask}}) = 1$), we have $\nabla_{\theta_g} \mathcal{L}_g^{\text{masked}} = \nabla_{\theta_g} \mathcal{L}_g$ as a mask ratio of '0' means no masking; **2)** if the masking ratio is set as '0.5' (*i.e.,* $p_g^{\text{mask}} = 0.5$ and $(1 - p_g^{\text{mask}}) = 0.5$), we have $\nabla_{\theta_g} \mathcal{L}_g^{\text{masked}} = 0.5 \nabla_{\theta_g} \mathcal{L}_g$ where the gradients are scaled down into half; **3)** if the masking ratio is set as '1.0' (*i.e.,* $p_g^{\text{mask}} = 1.0$ and $(1 - p_g^{\text{mask}}) = 0.0$), we have $\nabla_{\theta_g} \mathcal{L}_g^{\text{masked}} = 0.0 \times \nabla_{\theta_g} \mathcal{L}_g$ where the gradients are zeroized which means the generator will not be updated.

In practice, we consider updating GAN in mini-batches, where the generator update in conventional GAN can be defined as follow:

$$\theta_g^{m+1} = \theta_g^m + lr \times \nabla_{\theta_g} \mathcal{L}_g^m \tag{30}$$

where $lr$ denotes the learning rate for GAN training, *i.e.*, for both generator and discriminator training. $\theta_g^m$ stands for the generator parameters at $m$-th iteration while $\mathcal{L}_g^m$ denotes the generator training loss of $m$-th batch.

By Substituting Eq.29 into Eq.30, we have the generator update in MaskedGAN defined as:

$$\theta_g^{m+1} = \theta_g^m + lr \times \nabla_{\theta_g} \mathcal{L}_g^{\text{masked\_m}} \tag{31}$$

$$= \theta_g^m + lr \times (1 - p_g^{\text{mask}}) \nabla_{\theta_g} \mathcal{L}_g^m \tag{32}$$

$$= \theta_g^m + lr_g^{\text{mask}} \times \nabla_{\theta_g} \mathcal{L}_g^m, \tag{33}$$

where $lr_g^{\text{mask}} = lr \times (1 - p_g^{\text{mask}})$, and $\mathcal{L}_g^{\text{masked\_m}}$ denotes the generator training loss of $m$-th batch in MaskedGAN.

Following [8], let $\hat{h}(\theta_g, \theta_d) = h(\theta_g, \theta_d) + K_{(\theta_g)}$, where $\hat{h}(\theta_g, \theta_d)$ denotes a stochastic gradient of the generator loss function $\mathcal{L}_g$, $h(\theta_g, \theta_d)$ stands for the true gradient, $K_{(\theta_g)}$ denotes a random variable and $\theta_d$ denotes the parameters of the discriminator. Similarly, let $\hat{e}(\theta_g, \theta_d) = e(\theta_g, \theta_d) + K_{(\theta_d)}$ for the discriminator, where $\hat{e}(\theta_g, \theta_d)$, $e(\theta_g, \theta_d)$ and $K_{(\theta_d)}$ stands for the stochastic gradient, the true gradient and the random variable for discriminator, respectively. Then, we have the MaskedGAN update defined as follows:

$$\theta_d^{m+1} = \theta_d^m + lr \times (e(\theta_g^m, \theta_d^m) + K_{(\theta_d)}^m), \tag{34}$$

$$\theta_g^{m+1} = \theta_g^m + lr_g^{\text{mask}} \times (h(\theta_g^m, \theta_d^m) + K_{(\theta_g)}^m). \tag{35}$$

Since $lr_g^{\text{mask}} = lr \times (1 - p_g^{\text{mask}})$, when the masking ratio $p_g^{\text{mask}}$ approaches '1', we have $lr_g^{\text{mask}}(m) = o(lr(m))$, which satisfies the **Assumption 2** in [8], *i.e.*, the learning rate of the generator is far smaller than that of the discriminator. So far, we have proved that the MaskedGAN can be modeled as an instance of the Two Time-Scale Update Rule, where MaskedGAN can assign individual learning rates for discriminator and generator by using different masking ratios (*e.g.*, $p_g^{\text{mask}}$ for the generator while '0' for the discriminator).

Then, let us consider the convergence of the proposed MaskedGAN.

**Proposition 2** *The MaskedGAN converges to a Local Nash Equilibrium under certain conditions.*

Given the proofs (and the assumptions as well) in [8], Proposition 1 indicates that the proposed MaskedGAN converges to a Local Nash Equilibrium when the masking ratio $p_g^{\text{mask}}$ approaches '1'. So far, we have proved the convergence of the proposed MaskedGAN. In addition, the learning rate $lr$ used in Eq.34 could be replaced by an individual discriminator learning rate $lr_d^{\text{mask}}$ (with the masking ratio $p_d^{\text{mask}}$), which can prevent the discriminator from quickly becoming over-confident.

The convergence proof for training MaskedGAN assumes that the masking ratio used in generator training (*i.e.*, the masking ratio $p_g^{\text{mask}}$ in Proposition 1) will eventually become large enough (*i.e.*, $p_g^{\text{mask}}$ approaches '1') to ensure convergence of GAN. In practice, similar to the results in [8], we found that it is not necessary to set the masking ratio $p_g^{\text{mask}}$ as a very large value (*e.g.*, 0.75). The reason lies in that the actual learning pace of the generator is not only determined by the generator training masking ratio but also by generator loss function, generator architecture, generator optimization strategy, generator task difficulty, etc [8]. Therefore, the masking ratio used in generator training does not solely determine how fast the generator update is but serve to modulate it. Hence, in practice, the masking ratio used in generator training may be equal to the masking ratio used in discriminator training while making the generator changes slowly. These conclusions and analyses explain the rationality of the proposed MaskedGAN for training GANs with limited data.

## C   Empirical analysis of convergence

Section B provides theoretic insights to explain the rationality of the convergence of MaskedGAN, in which MaskedGAN is modelled as an example of the theory of Two Time-Scale Update Rule [8]. Among the conditions in Two Time-Scale Update Rule [8], the core (Assumption 2 in [8]) is to assign an individual learning rate for discriminator and generator such that GAN can converge to a local Nash equilibrium when the learning rate of generator is far smaller than that of discriminator

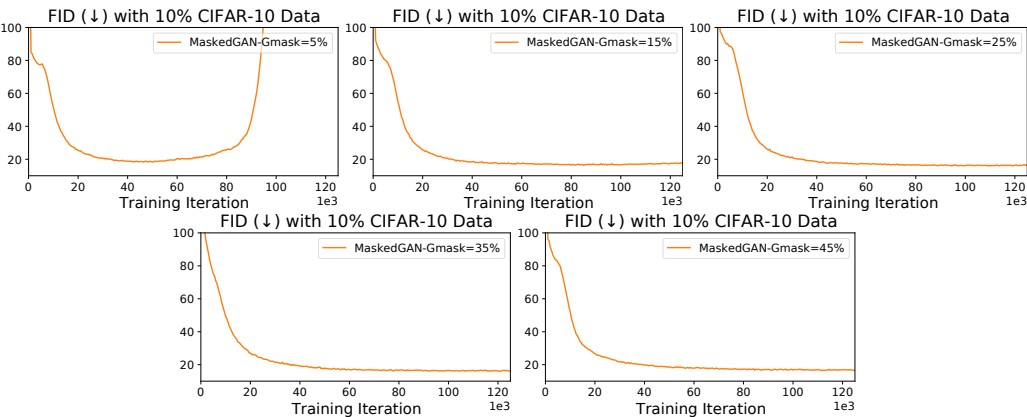

Figure 2: **Empirical analysis of MaskedGAN convergence** over BigGAN on 10% CIFAR-10 data. In MaskedGAN, the mask ratio in generator training controls the learning pace of generator by masking out a portion of training signals back-propagated from discriminator. As proved in [8], GAN converges to a local Nash equilibrium when the generator changes slowly enough. From this perspective, we analyze the MaskedGAN convergence empirically by changing the mask ratio in generator training, *i.e.*, adjusting the learning pace of generator (*e.g.*, higher mask ratios lead to faster learning pace). Specifically, we conduct experiments by keeping the spatial mask ratio in discriminator training unchanged (*i.e.*, 25%) while adjusting the spatial mask ratio in generator training (*e.g.*, 5%, 15%, 25%, 35% and 45%). It shows that 15%-45% mask ratios are large enough to ensure that the generator changes slowly enough while 5% is not enough to do so, where such empirical results are well-aligned with the theoretical insights.

| Spatial mask ratio in generator training | 5.0% | 15.0% | 25.0% | 35.0% | 45.0% |
|---|---|---|---|---|---|
| FID ($\downarrow$) | 18.20 | 16.30 | **15.89** | 16.20 | 16.40 |

Table 1: **Empirical analysis of MaskedGAN convergence** over BigGAN on 10% CIFAR-10 data. We provide the corresponding FID scores for the five models in Figure 2.

(i.*e.*, the generator changes slowly enough). Here, we study such theoretic insights by conducting practical experiments. In MaskedGAN, the mask ratio in generator training controls the learning pace of generator by masking out certain training signals back-propagated from discriminator, *e.g.*, '100%' mask ratio means no optimization by removing all training signals, '50%' indicates discarding half of the training signals and '0%' stands for training generator with all training signals. In this section, we conduct experiments by keeping the spatial mask ratio in discriminator training unchanged (*i.e.*, 25%) while adjusting the spatial mask ratio in generator training (*e.g.*, 5%, 15%, 25%, 35%, 45%). Figure 2 shows that 5% mask ratio works not bad (FID=~18) but still suffers from training divergence. On the other hand, 15%-45% mask ratios achieve better performance (FID=~15.9) and ensure good convergence of MaskedGAN under limited training data. These experimental results reveal that 15%-45% mask ratios are large enough to ensure that the generator changes slowly enough while 5% is not enough to do so, where such empirical results are well-aligned with the theoretical insights. In addition, note that the performance and convergence degradation of 5% mask ratio is not caused by the "different objectives" issue illustrated in DA [18], because other different masking ratios (*e.g.*, 15%, 35%, 45%) work and converge well. This also indicates that image masking has very different properties as compared with conventional data augmentation [18].

# D    Experiment details

## D.1    Datasets

**CIFAR-10 and CIFAR-100:** They both contain $50,000$ samples in training set and $10,000$ samples in validation set, where the image resolutions are $32 \times 32$.

**ImageNet:** ImageNet spans 1000 object classes and contains $1,281,167$ training images. We use the resolutions $64 \times 64$ in experiments.

**100-shot:** 100-shot contains three datasets with 100 training samples for each dataset: 100-shot Obama, 100-shot Grumpy Cat, 100-shot Panda, where the image resolutions are $256 \times 256$.

**AFHQ:** AFHQ is a dataset of animal faces consisting of three domains (*i.e.* cat, dog, and wildlife), each domain provides 5,000 training images. We use the resolutions $256 \times 256$ in experiments.

**FFHQ:** FFHQ is a high-quality image dataset of human faces and it contains $70,000$ training images. We use the resolutions $256 \times 256$ in experiments.

**Cityscapes:** Cityscapes is a real-world street image dataset with $24,998$ training images. We use the resolutions $256 \times 256$ in experiments.

## D.2  Implementation Details

**BigGAN on CIFAR-10 and CIFAR-100:** We build our MaskedGAN (*i.e.*, Masked-BigGAN) on top of the PyTorch implementation of BigGAN in [18]. We set a learning rate of $2e-4$ for both $G$ and $D$, a batch size of 50 and use Adam optimizers with $\beta_1 = 0$, $\beta_2 = 0.999$, and $\epsilon = 10^{-8}$. The FID evaluation is performed on the whole validation set.

**BigGAN on ImageNet:** We implement our MaskedGAN (*i.e.*, Masked-BigGAN) based on the PyTorch version of BigGAN in [18]. Following [18], we use a learning rate of $1e-4$ for G and $4e-4$ for D for 100% data setting and reduce the learning rate of D to $2e-4$ for the 5% and 2.5% data settings. We set batch size of 512 and use Adam optimizers with $\beta_1 = 0$, $\beta_2 = 0.999$, and $\epsilon = 10^{-8}$. The FID evaluation is performed on the whole training set.

**StyleGAN-v2 on 100-shot, AFHQ and FFHQ:** We build our MaskedGAN (*i.e.*, Masked-StyleGAN-v2) on top of PyTorch implementation of StyleGAN-v2 in [18]. Following [18], we use a learning rate of $2e-3$ for G and D. We set batch size of 8 and use Adam optimizers with $\beta_1 = 0$, $\beta_2 = 0.99$, and $\epsilon = 10^{-8}$. The FID evaluation is performed on the whole training set.

**TransGAN on CIFAR-10 and CIFAR-100:** We build our MaskedGAN (*i.e.*, Masked-TransGAN) on top of PyTorch implementation of TransGAN [10]. Same as TransGAN, we adopt a learning rate of $1e-4$ for both G and D, a batch size of 128 for generator and 64 for discriminator and use Adam optimizers with $\beta_1 = 0$, $\beta_2 = 0.99$, and $\epsilon = 10^{-8}$. The FID evaluation is performed on the whole training set.

**GANformer on Cityscapes:** We build our MaskedGAN (*i.e.*, Masked-GANformer) on top of PyTorch implementation of GANformer [9]. We adopt a learning rate of $1e-3$ for both G and D, a batch size of 32 and use Adam optimizers with $\beta_1 = 0$, $\beta_2 = 0.99$, and $\epsilon = 10^{-8}$. The FID evaluation is performed on the whole training set.

We set the mask ratio as 25.0% for both 'shifted spatial masking' and 'balanced spectral masking', the mask patch size as $8 \times 8$, the number of decomposed spectral bands as 64. Please refer to Section E for parameter ablation experiments. Note the mask patch size is scaled linearly when the image size changes, *i.e.*, we double the mask patch size if image size doubles. To avoid the training conflict between samples generated by spatial masking and spectrum masking, we employ two individual discriminators for them. We conduct experiments in Pytorch with Tesla V100 GPUs.

## E  Parameter ablations

In this section, we conduct parameter ablations for the proposed MaskedGAN:

**(a) Mask ratio in shifted spatial masking:** Shifted spatial masking randomly masks out a portion of image patches in spatial dimensions, where the mask ratio controls the percentage of the masked content. We studied the mask ratio (in shifted spatial masking) by changing it from 0% to 50% with a step of 12.5%. Table 2a reports the experimental results with with BigGAN on 10% CIFAR-10 data. It shows that the spatial mask ratio does not affect the generation performance clearly while it changes from 12.5% to 37.5%. When the spatial mask ratio increases into 50%, the generation performance drops a bit. It is reasonable as a large spatial mask ratio may make the whole framework becomes hard-to-optimize.

| Mask ratio | 0% | 12.5% | 25.0% | 37.5% | 50.0% |
|---|---|---|---|---|---|
| FID (↓) | 28.78 | 16.50 | **15.89** | 16.70 | 18.90 |

(a) **Mask ratio in shifted spatial masking.**

| Mask patch size | $6 \times 6$ | $7 \times 7$ | $8 \times 8$ | $9 \times 9$ | $10 \times 10$ |
|---|---|---|---|---|---|
| FID (↓) | 16.25 | 16.94 | **15.89** | 15.98 | 16.41 |

(b) **Mask patch size in shifted spatial masking.**

| Mask type | Random patches | Random blocks | Random squares | Center Grid |
|---|---|---|---|---|
| FID (↓) | **15.89** | 17.85 | 17.10 | 39.54 |

(c) **Mask type in shifted spatial masking.**

| Mask ratio | 0% | 12.5% | 25.0% | 37.5% | 50.0% |
|---|---|---|---|---|---|
| FID (↓) | 22.56 | 16.19 | **15.89** | 19.00 | 19.02 |

(d) **Mask ratio in balanced spectral masking.**

| Spectral bands | 32 | 48 | 64 | 80 | 96 |
|---|---|---|---|---|---|
| FID (↓) | 16.59 | 16.06 | **15.89** | 15.88 | 15.86 |

(e) **The number of decomposed spectral bands in balanced spectral masking.**

Table 2: **MaskedGAN parameter ablation experiments** with BigGAN on 10% CIFAR-10 data. We report FID (↓) scores. If not specified, the default is: in *shifted spatial masking*, the mask ratio is 25%, the mask patch size is $8 \times 8$, the mask type is 'random patches'; in *balanced spectral masking*, the mask ratio is 25%, the number of decomposed spectral bands is 64. Default settings are marked in gray .

**(b) Mask patch size in shifted spatial masking:** Shifted spatial masking randomly masks out a portion of image patches in spatial dimensions, where the mask patch size controls the size of mask unit. We studied the mask patch size (in shifted spatial masking) by changing it from $6 \times 6$ to $10 \times 10$ with a step of $1 \times 1$. Table 2b reports the experimental results with with BigGAN on 10% CIFAR-10 data. We can observe that the image generation performance is not sensitive to the mask patch size.

**(c) Mask type in shifted spatial masking:** Shifted spatial masking randomly masks out a portion of image in spatial dimensions, where the mask type affects the performance. We studied the mask type (in shifted spatial masking) by using four different masks as in [7], such as random patches, random blocks, random squares and center grid. Table 2a reports the experimental results with BigGAN on 10% CIFAR-10 data. It shows that masking images uniformly randomly works well (*e.g.*, Random patches, Random blocks and Random squares) , in which 'random patches' works the best. On the contrary, the static masking, *i.e.*, center grid masking that masks the center in each image grid, does not work well.

**(d) Mask ratio in balanced spectral masking:** Balanced spectral masking masks out certain image spectral bands in the spectral dimension, where the mask ratio controls the percentage of the masked content. We studied the mask ratio (in balanced spatial masking) by changing it from 0% to 50% with a step of 12.5%. Table 2a reports the experimental results with with BigGAN on 10% CIFAR-10 data. It shows that the spectral mask ratio does not affect the generation performance clearly while it changes from 12.5% to 25.0%. When the spectral mask ratio increases to 37.5%-50%, the generation performance drops a bit. It is reasonable as a large spectral mask ratio may remove too much image information and make the whole framework becomes hard-to-optimize.

**(e) The number of decomposed spectral bands in balanced spectral masking:** Balanced spectral masking masks out certain image spectral bands in the spectral dimension, where the number of decomposed spectral bands controls the resolution in the spectral dimension. We studied the number of decomposed spectral bands (in balanced spatial masking) by changing it from 32 to 64 with a step

Table 3: Unconditional image generation with TransGAN on CIFAR-10 and CIFAR-100 over 100% training data. MaskedGAN works effectively with transformer-based GAN (TransGAN) as well. We calculate FID ($\downarrow$) scores with 10K generated samples and the training set.

| Method | CIFAR-10 (FID$\downarrow$) | CIFAR-100 (FID$\downarrow$) |
|---|---|---|
| WGAN-GP [6] | 39.68 | - |
| AutoGAN [5] | 12.42 | - |
| AdversariaAS-GAN [5] | 10.87 | - |
| Progressive-GAN [11] | 15.52 | - |
| **TransGAN** [10] | 22.53 | - |
| **TransGAN** + DA [10] | 9.26 | - |
| **TransGAN** (reproduced) (baseline) | 21.44 | 25.73 |
| **TransGAN** + DA (reproduced) | 9.30 | 17.70 |
| **MaskedGAN** | **8.71** | **16.88** |

of 16. Table 2a reports the experimental results with with BigGAN on 10% CIFAR-10 data. It shows that the number of decomposed spectral bands does not affect the generation performance clearly as long as it is large enough, *e.g.*, 48-96.

## F   Comparisons with DA [18] over TransGAN

We note that TransGAN [10] studied data augmentation for transformer-based GAN by slightly adjusting and employing DA [18]. In this section, we compare MaskedGAN with DA [18] over transformer-based GAN. Table 3 reports the results over TranGAN on CIFAR-10 and CIFAR-100 with 100% training data. It shows that our MaskedGAN outperforms DA [18] over transformer-based GAN as well.

## G   Comparisons with the masking strategy used in MAE

In this section, we compare MaskedGAN with the masking strategy used in Masked Autoencoders [7] (MAE). Table 4 reports the results with BigGAN (baseline) on 10% CIFAR-10 data. It shows that directly applying the MAE masking strategy [7] on GAN (the 'spatial masking' in Table 4) could improve the performance in some degree (*i.e.*, achieves ~28 in FID) but cannot achieve state-of-the-art performance. The major reason is that these methods were designed for unsupervised representation learning without considering generating or reconstructing high-fidelity images. As a comparison, with the proposed masking strategies, our MaskedGAN (the second last row in Table 4) produces better performance clearly (*i.e.*, achieves ~15.8 in FID) and defines new state-of-the-arts for training GAN with limited data. This is largely because MaskedGAN tailors several masking strategies for GAN, such as 'mask shift' for spatial masking and 'balanced spectral masking', which are essential to high-fidelity image generation.

## H   Comparisons with the 'cutout' used in ADA [12] and DA [18]

We note that ADA [12] and DA [18] introduced massive data augmentation techniques from visual recognition into GAN training, such as color transformations, adding noises, cutout, etc. Among these data augmentations, the 'cutout' operation looks similar to the basic *spatial masking* used in MAE and our MaskedGAN. In this section, we conduct experiments to study the relation between the basic *spatial masking* (used in MAE and our MaskedGAN) and the 'cutout'. Table 4 reports the results over BigGAN (baseline) on 10% CIFAR-10 training data. It shows that including the 'cutout' operation produces a similar but a little bit lower performance as compared with the basic *spatial masking* (used in MAE and our MaskedGAN), indicating that the basic *spatial masking* (*i.e.*, 'random patches') is more suitable than the 'cutout' for image generation. The major reason is that the basic *spatial masking* (*i.e.*, 'random patches' used in MAE and our MaskedGAN) can mask out each

Table 4: Comparisons with the 'cutout' used in ADA [12] and DA [18] with BigGAN (baseline) on 10% CIFAR-10 data.

| Method | Shifted Spatial Masking | | Balanced Spectral Masking | | FID ($\downarrow$) |
|---|---|---|---|---|---|
| | Spatial Masking | Random Shift | Spectral Masking | Self-adaptive Probability | |
| BigGAN [2] | | | | | 48.08 |
| Random patches in MAE [7] | ✓ | | | | 28.35 |
| | ✓ | ✓ | | | 22.56 |
| | | | ✓ | | 35.20 |
| | | | ✓ | ✓ | 28.78 |
| **MaskedGAN** | ✓ | ✓ | ✓ | ✓ | 15.89 |
| Cutout in [12, 18] | | | | | 29.52 |

image locations/pixels with uniform probabilities and a fixed overall mask ratio, ultimately leading to unbiased image masking and a stable training process. As a comparison, the 'cutout' operation cannot achieve the above two properties simultaneously. *First*, if we conduct 'cutout' without padding, it masks the center locations/pixels with higher probabilities while masking the edge locations/pixels with lower probabilities, leading to biased image masking and degraded image generation. *Second*, if we conduct 'cutout' with padding, it can mask every location with the same probability but its mask ratio is not fixed and unstable, *i.e.*, the overall mask ratio decreases from $p^{mask}$ (the ratio between the cutout size and the whole image) to '0' when the cutout mask moves from center locations to corner locations, which may make the overall training unstable and lead to degraded performance.

Nevertheless, the results in Table 4 show that using pure spatial masking only could improve the performance in some degree but cannot achieve state-of-the-art performance. In MaskedGAN, further including the proposed 'random mask shift' and 'balanced spectral masking' improves the performance clearly and defines new state-of-the-art for training GAN with limited data.

In addition, the results in Table 4 also show that MaskedGAN achieves state-of-the-art performance without using the massive data augmentations such as color transformation and noises addition (as in ADA [12] and DA [18]). It reveals that the key point of training GAN with limited data is preventing trivial solutions, *e.g.*, encouraging the networks to learn useful and holistic knowledge, which can be achieved by simple masking strategies. These results are consistent with the conclusions in [7] which shows that the task is made difficult by image masking and thus requires less augmentation to regularize training.

# I  Relations to other so-called MaskedGAN

We note that there are two works that are also named 'MaskedGAN' [17, 13]. [17] is for unsupervised monocular depth and ego-motion estimation, which uses the generator to produce a 'Boolean mask' to perform the same occlusions of synthesised images on the target images, aiming for eliminating the effects of occlusions in unsupervised monocular depth and ego-motion estimation. [13] is for Face Editing, which uses semantic masks (generated by DeepLabV3) to select the regions for editing. Obviously, these two works are quiet different to this paper, *e.g.*, in perspectives of topic/task and the concept of 'mask'. Besides, we also note that there is a very new work named "Maskgit: Masked generative image transformer" [3]. [3] is based on VQVAE [15], which generates images with pre-trained codebooks in two stages: 1) it uses a pre-trained model (*e.g.*, such as VQVAE [15] and DALL-E [14]) to encode images into "visual words"; 2) it trains an autoregressive model (*e.g.*, transformer) to generate image tokens (or "visual words") sequentially according to the previously generated ones (*i.e.* autoregressive decoding). [3] proposes a bidirectional transformer decoder named MaskGIT that learns to predict randomly masked tokens by attending to tokens in all directions, which could improve the second stage by refining the image iteratively conditioned on the previous generated results. Clearly, [3] is quiet different to this paper, where we focus on training GAN with limited data while [3] focuses on improving the second stage of VQVAE-based generative models without any considerations on GAN or data-limited regime. This statement is consistent with the claims in [3], *i.e.*, "unlike the subtle min-max optimization used in GANs, VQVAE-based generative models are learned by maximum likelihood estimation".

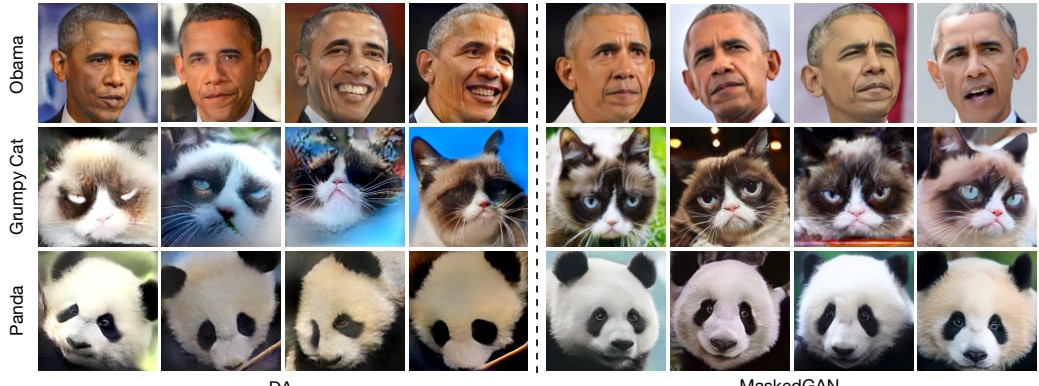

Figure 3: **Qualitative illustration and comparison** over 100-shot Obama, Grumpy Cat and Panda: With limited training data, introducing the proposed image masking strategies into GAN training helps generate more realistic and high-fidelity images, especially in terms of image shapes and structures.

## J   More qualitative illustrations

We provide more qualitative illustrations in Fig. 3, which show that MaskedGAN outperforms the baseline and state-of-the-art models clearly. Specifically, with limited training data, introducing the proposed image masking strategies into GAN training helps generate more realistic and high-fidelity images, especially in terms of image shapes and structures.