# OpenReview forum: "Masked Generative Adversarial Networks are Data-Efficient Generation Learners"
_NeurIPS.cc/2022/Conference — NeurIPS 2022 Accept_

### Official Review · Reviewer_S4RA · 2022-07-11

**Rating:** 6
**Confidence:** 4
**Soundness:** 3 good
**Presentation:** 3 good
**Contribution:** 3 good

**Summary:**

This paper proposes MaskedGAN to deal with the problem of training GAN with limited data. MaskedGAN uses two masking strategies, shifted spatial masking (masking spatial patches) and balanced spectral masking (masking spectral bands), on both real images and generated images during the training of GAN. Experimental results shows the effectiveness of the proposed method over the original GAN, as well as the superiority over other techniques for training GAN with limited data, across different architectures and datasets.

**Questions:**

1. My major concern is why the comparison results with other techniques for training GAN with limited data on AFHQ, FFHQ, and ImageNet are missing. The experimental results are not convincing enough without them.

2. Masking, including spatial masking and spectral masking, can also be viewed as a kind of data augmentation strategy. What makes masking, or the specially designed masking in this paper, superior to other data augmentation strategies?

3. How much does the computation overhead increase after adding the masking strategy during GAN training?

**Limitations:**

The authors did not state the limitation of the proposed method clearly in the paper. They only claimed possible future applications, e.g., applying MaskedGAN to multi-modality generation, but those are not the limitation of the method or result itself.

**Strengths And Weaknesses:**

Pros:
1. The idea of developing masking strategies to enhance the training of GAN with limited data is great. It shows the power of masking in the generative field.

2. The results are strong. This paper tests the proposed method with different architectures and datasets, and shows consistent and remarkable improvements over baselines, especially when the ratio of available data is small.

Cons:

In the experiments, the proposed method compares with other techniques for training GAN with limited data (e.g., DA, GenCo, etc) only on very small datasets (e.g., CIFAR, 100-shot) but not on large datasets (e.g., ImageNet, FFHQ). On large datasets, it only compares with the baseline.

---

> ### Author Response · Authors · 2022-08-02
> **Thank you for your confirmation of the value of developing masking strategies for data-efficient GAN and the strong experimental results. Below please find our responses regarding your concerns. Part 1 (Q1)**
>
> Q1: My major concern is why the comparison results with other techniques for training GAN with limited data on AFHQ, FFHQ, and ImageNet are missing. The experimental results are not convincing enough without them?
> - Thank you for the suggestion! We conducted the suggested experimental comparisons over ImageNet.
> - We would clarify that we did not benchmark with ADA-related methods including InsGen and APA as our work mainly follows DA that adopts different experiment setups. To benchmark with ADA-related methods, it requires to rerun their codes over DA's experiment setups for valid comparisons. Such experiments are super computational intensive, e.g., ADA takes approximately 1, 259, 337.6 GPU hours (i.e., 143.76 GPU years) to complete its experiments. We did not have sufficient GPU resources to conduct such experiments.
> - Therefore, due to the computation resource constraints, we only benchmarked DA and ADA over the very large ImageNet dataset, the very small 100-shot dataset and the medium-size CIFAR-10/100 dataset, where we believe extensive experiments over these datasets are sufficient for benchmarking the proposed method. We plan to conduct related experiments and benchmark DA and ADA over other datasets and backbones later.
> - In addition, the experiments and comparisons over the 100-shot dataset (i.e., the revised Table 4) are very relevant and meaningful as this paper focuses on training GAN with limited data, where the task would be more challenging while working with a small dataset.
>
> Revised Table 3: Conditional image generation with BigGAN on ImageNet (FID).
> | Method                          | 10% Data | 5% Data  | 2.5% Data |
> | --------------------------------|:-------:|:-----:|:-----:|
> |BigGAN (baseline)    |38.30  ±  0.25   |91.16  ±  0.43  |133.80  ±  0.76
> |ADA (**newly included**)   | 31.89  ±0.17 |43.21   ±  0.37  | 56.83   ±  0.48
> |DA (**newly included**)    | 32.82    ±  0.18  |56.75   ±  0.35 |63.49   ±  0.51
> |**MaskedGAN**   | 26.51  ±  0.12  | 35.70  ±  0.31   | 38.62  ±  0.37
>
> Revised Table 3: Conditional image generation with BigGAN on ImageNet (IS).
> | Method                          | 10% Data | 5% Data  | 2.5% Data |
> | --------------------------------|:-------:|:-----:|:-----:|
> |BigGAN (baseline)  |10.94  ±  0.35 |6.13  ±  0.09  |3.92  ±  0.07  |
> |ADA (**newly included**)  |12.67  ±0.31   |9.44  ±0.25   |8.54  ±   0.26  |
> |DA (**newly included**)  |12.76  ±  0.34    |9.63  ±  0.21  |8.17  ±  0.28 |
> |**MaskedGAN**   |13.34  ±  0.24   | 12.85  ±  0.40  | 12.68  ±  0.27  |
>
> Revised Table 1: Conditional image generation with BigGAN on CIFAR-10 (FID).
> | Method                          | 100% Data | 20% Data  | 10% Data |
> | --------------------------------|:-------:|:-----:|:-----:|
> | Non-saturated GAN |9.83  ±  0.06  |18.59 ±  0.15   |41.99 ±  0.18 |
> | LS-GAN |9.07 ±  0.01 | 21.60 ±  0.11 | 41.68 ±  0.18
> | RAHinge GAN | 11.31 ±  0.04 | 23.90 ±  0.22| 48.13 ±  0.33|
> | StyleGAN-v2 | 11.07 ±  0.03  | 23.08 ±  0.11  | 36.02 ±  0.15  |
> | BigGAN (baseline)  | 9.74 ±  0.06| 21.86 ±  0.29| 48.08 ±  0.10 |
> | LeCam-GAN| 8.31 ±  0.05 |15.27 ±  0.10 |35.23 ±  0.14 |
> | GenCo | 8.83 ±  0.04 | 16.57 ±  0.08 |  28.08 ±  0.11|
> | ADA (**newly included**) |8.99  ±  0.03 |19.87  ±  0.09 |30.58  ±  0.11 |
> | DA | 8.75 ±  0.03 |  14.53 ±  0.10| 23.34 ±  0.09 |
> | **MaskedGAN**  | 8.41 ±  0.03 | 12.51 ±  0.09| 15.89  ±  0.12 |
>
> Revised Table 1: Conditional image generation with BigGAN on CIFAR-100 (FID).
> | Method                          | 100% Data | 20% Data  | 10% Data |
> | --------------------------------|:-------:|:-----:|:-----:|
> | Non-saturated GAN | 13.87 ±  0.08|  32.64  ±  0.19 |  70.5 ±  0.38|
> | LS-GAN|  12.43 ±  0.11|  27.09 ±  0.09|  54.69 ±  0.12  |
> | RAHinge GAN| 14.61 ±  0.21| 28.79 ±  0.17|  52.72 ±  0.18|
> | StyleGAN-v2| 16.54 ±  0.04 |  32.30  ±  0.11|  45.87 ±  0.15 |
> | BigGAN (baseline) | 13.60 ±  0.07|  32.99 ±  0.24| 66.71 ±  0.01|
> | LeCam-GAN | 11.88 ±  0.12| 25.51 ±  0.19| 49.63 ±  0.16|
> | GenCo|   11.90 ±  0.02| 26.15} ±  0.08| 40.98 ±  0.09|
> | ADA (**newly included**) | 12.22  ±  0.02|  22.65  ±  0.10| 27.08  ±  0.15|
> | DA|   11.99 ±  0.02|22.55 ±  0.06|35.39 ±  0.08 |
> | **MaskedGAN** |   11.65 ±  0.03|  18.33 ±  0.09| 24.02 ±  0.12|
>
> Revised Table 4: Unconditional image generation with StyleGAN-v2 on 100-shot dataset (FID).
> | Method                          | Obama | Grumpy Cat  | Panda |
> | --------------------------------|:-------:|:-----:|:-----:|
> StyleGAN-v2  (baseline) | 80.20  | 48.90  | 34.27
> ADA  | 45.69  | 26.62 | 12.90
> LeCam-GAN  | 38.58  | 41.38  | 19.88
> GenCo  | 36.35  | 33.57 |  15.50
> AdvAug  | 52.86  | 31.02  | 14.75
> DA  |46.87 | 27.08 | 12.06
> APA (**newly included**)  | 43.75  | 28.49  |  12.34
> InsGen (**newly included**)  | 45.85  | 27.48  | 12.13
> **MaskedGAN**   | 33.78 ± 0.27  | 20.06 ± 0.13  | 8.93 ± 0.06

---

> > ### Author Response · Authors · 2022-08-02
> > **Part 2 (Q2-Q3)**
> >
> > Q2: Masking, including spatial masking and spectral masking, can also be viewed as a kind of data augmentation strategy. What makes masking, or the specially designed masking in this paper, superior to other data augmentation strategies?
> > - Thank you for your questions. We would clarify that we discussed the difference between our MaskedGAN and data augmentation methods extensively in our manuscript and appendix. For example, the 4th paragraph of Section 4.4 describe the better convergence of the proposed MaskedGAN and why it converges better than existing data augmentation methods; Section 3.3 provides detailed theoretical insights and illustrations (on two time-scale update rule and local Nash Equilibrium); Section H of appendix shows the comparison with "cutout" that is used in data augmentation methods DA and ADA.
> > - In summary, MaskedGAN designs two image masking strategies that work by removing certain image information only while previous data augmentation methods involve various data augmentations such as color jitters, saturation adjustment, etc. Besides, different to the conventional "cutout’ in DA and ADA, MaskedGAN uses "random patch-based masking" and designs "random mask shift" and "balanced spectral masking".
> > Such differences in designs lead to very different results in convergences (empirically and theoretically) and generation performances.
> > More detailed descriptions can found in the following texts. Thank you for your suggestion and we will include the above discussions in the revised manuscript.
> >
> >
> > - Followed please find the detailed texts copied (or summarized) from our manuscript and appendix for your reference, which extensively discuss the difference between our MaskedGAN and data augmentation methods:
> >
> > - **1)** As mentioned in the 4th paragraph ("Convergence comparison across different network architectures and datasets") of Subsection 4.4, the experimental results show that MaskedGAN converges well consistently across various conditions (the amounts of training data, network architectures and datasets) while the data augmentation method such DA still suffers from generation failures and training collapses. The great convergence of MaskedGAN is largely attributed to two factors: (a) its image masking designs suppress trivial solutions and training failures directly; (b) it keeps similar learning paces for discriminator and generator which ensures that network converge to a Local Nash Equilibrium under certain conditions [21].
> > MaskedGAN can achieve factor (b) as its image masking strategies (or called data augmentation operations) operate by masking (or more specifically, removing) some image information only.
> > Differently, data augmentation methods such as DA and ADA cannot guarantee factor (b) as they generally includes some operations like color jitters and saturation adjustment which cannot satisfy the theoretic proofs as introduced in Section 3.3.
> >
> > - **2)** Point 1) described the design differences between MaskedGAN and previous data augmentation methods (e.g., DA and ADA) and illustrated different designs lead to different empirical convergences and further different generation performances.
> > In Section 3.3, we provided detailed theoretic insights and illustrations, which show MaskedGAN can be modeled as an instance of Two Time-Scale Update Rule and and thus it converges to a Local Nash Equilibrium under certain conditions.
> > On the other hand, previous data augmentation methods (e.g., DA and ADA) cannot satisfy our Propositions 1 and 2 provided in Section 3.3 because data augmentation methods such as DA and ADA includes some operations like color jitters and saturation adjustment, where such additive noises violate the proposition condition of removing certain image information only.
> >
> > - **3)** In Section H in appendix ("Comparisons with the ‘cutout’ used in ADA and DA"), we compared our MaskedGAN with the ‘cutout’ used in ADA and DA by providing experiments and detailed explanations.
> >
> > Q3: How much does the computation overhead increase after adding the masking strategy during GAN training?
> > - The masking strategy could be considered as a type of data augmentation, which includes the Fourier transformation, inverse Fourier transformation and zeroing operations that introduce little extra computation overhead (similar to the previous works ADA and DA).

---

> > > ### Comment · Reviewer_S4RA · 2022-08-09
> > > **Thanks for the response**
> > >
> > > Thanks for the additional experiments and analysis provided by the authors. My concerns are well addressed. I will raise my score.

---

### Official Review · Reviewer_mM3P · 2022-07-11

**Rating:** 3
**Confidence:** 5
**Soundness:** 1 poor
**Presentation:** 1 poor
**Contribution:** 3 good

**Summary:**

This work suggests two strategies for masking in GANS and it claims that their Mask Generative Adversarial Networks (MaskedGAN) are robust with limited training data.
The two strategies are:

1) Shifted spatial masking (random shifts in the spatial domain)
2) Balanced spectral masking (multiple bands with self-adaptive probabilities)

Hypothesis: masking helps the model to learn hard-to-discriminate band and create a challenging scenario
Support for the hypothesis: training on multiple architectures (not covering all datasets)
Limited dataset setup:
10%(5K), 20%(10K), and 100%(50K) of data on CIFAR10, 100
2.5%(25K), 5%(50K), and 10%(100K) of data on ImageNet



**Questions:**

- What do you mean by Masking? Are you zeroing some pixels? In that case, are you using Gated or Partial Convolutions? If yes, then the models are different, and if no, how do you handle the normalization of the output to adjust for the fraction of missing data?
- Is the sampled data heterogeneous or homogeneous? How have you handled it?
- What are the learning rate, batch-size, and epochs of training for each of the experiments?
- In Fig2, I don't understand how the masked image is used in the model? Specifically, can you make it clear if the model receives the right images directly? If it receives the lower image (Masked Image in the spectral domain), it seems a completely different image! Do some of the pixels retain the original pixel values?


**Limitations:**

No. They mentioned the limited data training, which is the main setting of the paper, but as far as I understand, listing the limitation means to discuss where the methodology does not work.

**Strengths And Weaknesses:**

Strengths:
- The paper introduces an interesting subject for the community.
- There are multiple experiments.
- Multiple models are used for the experiments.

Weaknesses:
- The writing is somewhat incoherent and unclear: in the abstract and introduction, problem statements lack motivation and clarity.
- The literature review section does not provide coherent statements for each cited work, and it bulk references some papers that might not be directly related. e.g. for Masked Autoencoders, it references basic papers of autoencoders(45, 39, 8, 17, 2, 20, 48), which needs to be discussed in the Autoencoder section.
- The mentioned shifting and Fourier domain manipulation (masking) are not clearly discussed.
- It is not clear how the sampling happens in the limited data scenario and if it is heterogeneous or homogenous.
- The number of training rounds is not specified.
- It seems it works as an augmentation, and with multiple random masking, we actually increase the number of data samples. I think after mentioning the number of iterations, it would be appropriate to compare with SOTA augmentation techniques in GANs.

---

> ### Author Response · Authors · 2022-08-02
> **Thank you for your appreciation of our introduction of an interesting subject for the community and extensive experiments over multiple datasets and models. Below please find our responses regarding your concerns. Part 1 (Q1-Q4)**
>
> Q1: The literature review section does not provide coherent statements for each cited work, and it bulk references some papers that might not be directly related. e.g. for Masked Autoencoders, it references basic papers of autoencoders(45, 39, 8, 17, 2, 20, 48), which needs to be discussed in the Autoencoder section?
> - We would clarify that the references [45, 39, 8, 17, 2, 20, 48] have no problem and all of them propose new Masked Autoencoders (MAE) with different image masking strategies. For example, [45] is the pioneering work of MAE, which presents masking as a noise type in Denoising Autoencoders (DAE); Context Encoder [39] proposes to masks random image regions of different shapes. iGPT [8], ViT [17], BEiT [2], MAE [20] and SimMIM[48] are inspired by transformers [17]: [8] proposes masked pixel prediction; [17] uses masked patch prediction; based on [17], [2] proposes the tokenization with a block-wise masking strategy, [20] proposes to use a high masking ratio and [48] proposes several simple designs (e.g., a large masked patch size and a light prediction head). Thus, we believe we provided coherent statements for each cited work in the section of Related Work.
> - Note the recent well-known Masked Autoencoders papers [20, 48] also cited [45, 39, 8, 17, 2] as the related works of Masked Autoencoders (or called "Masked image encoding/modeling"), i.e., the third paragraph of Section 2 in [20] and the second paragraph of Section 2 in [48].
>
> Q2: The mentioned shifting and Fourier domain manipulation (masking) are not clearly discussed?
> - We would clarify that we clearly defined the "random shift" (Lines138-150) and "Fourier domain manipulation (masking)" (Lines159-175) in Section 3.2. We also extensively discussed these two designs in our manuscript and appendix. Following please find the detailed texts copied (or summarized) from our manuscript and appendix for your reference:
>
> - In Table 2, we presented ablation studies to discuss and examine how each design (i.e., "Random Shift", "Spectral Masking" and "Self-adaptive Probability") contributes to the overall performance (Lines207-221).
>
> - In Section E in appendix ("Parameter ablations"), we presented parameter studies which extensively discussed the parameters used in MaskedGAN (including the parameter involved in these two designs).
>
> Q3: About implementation details. It is not clear how the sampling happens in the limited data scenario and if it is heterogeneous or homogenous? Is the sampled data heterogeneous or homogeneous? How have you handled it? The number of training rounds is not specified? What are the learning rate, batch-size, and epochs of training for each of the experiments?
> - We would clarify that we adopted similar training details as in earlier data augmentation methods such as DA and ADA.
> We provided the experiment detail in Second D in appendix ("Experiment details"), which includes the detail information of datasets, backbones, network training, as well as the training details such as the learning rate, the batch-size and other specifics used for each backbone and dataset.
> -  We adopted the same data sampling random seed (i.e., selecting which data for limited the data scenario) and number of training rounds as used in the previous work DA.
> We did not provide these information as most data-efficient GAN studies follow DA (or ADA) to conduct the experiments and all the training details (e.g., the random seed of data sampling, the number of training rounds, the learning rate, etc.) are the same as used in DA (or ADA).
> -  As mentioned, we followed DA to conduct the experiments, where the data sampling is heterogeneous (e.g., if the full training data are evenly distributed across categories, after sampling, the number of training data may be different for each category).
>
> Q4: It seems it works as an augmentation, and with multiple random masking, we actually increase the number of data samples. I think after mentioning the number of iterations, it would be appropriate to compare with SOTA augmentation techniques in GANs?
> - As mentioned in the response to the previous question (i.e., Q4 from Reviewer mM3P), we followed DA to conduct the experiments, where the number of training epochs is the same as in DA.

---

> > ### Author Response · Authors · 2022-08-02
> > **Part 2 (Q5-Q6)**
> >
> > Q5: What do you mean by Masking? Are you zeroing some pixels? In that case, are you using Gated or Partial Convolutions? If yes, then the models are different, and if no, how do you handle the normalization of the output to adjust for the fraction of missing data?
> > - As mentioned in Section 3.2 (Lines138-150 and Lines159-175), image masking is defined as multiplying the image with a binary mask (e.g., $M_{spatial}(x) = x \times m_{spatial}'$ where $m_{spatial} \in$ {0,1} $^{H \times W}$). This definition is the same as in Masked Autoencoders papers (e.g., Context Encoder [39], MAE [20], etc.).
> >
> > - We did not use Gated or Partial Convolutions.
> > - We conducted the image masking operations after image normalization during training. The Masked Autoencoders papers (e.g., Context Encoder [39], MAE [20], etc.) also conducted image masking operations in the similar way and all of them show that such image masking operations have no problem and work well over both CNNs-based and Transformer-based networks. We reckon this should not be a problem as none of these works (e.g., Context Encoder [39], MAE [20], etc.) reported any normalization issues of zeroing some pixels in CNNs-based and Transformer-based network training.
> >
> > Q6: In Fig2, I don't understand how the masked image is used in the model? Specifically, can you make it clear if the model receives the right images directly? If it receives the lower image (Masked Image in the spectral domain), it seems a completely different image! Do some of the pixels retain the original pixel values?
> > - In Fig 2, both "Masked Image" are directly fed into the networks for training. We confirmed that the model receives the right images (i.e., the two Masked Image on the right) directly.
> >
> > - For the second "Masked Image" at the bottom in Fig 2 (Masked Image in the spectral domain), it seems quite different to the original image as the low-frequency bands have been masked (i.e., removed). If we zoom in it, we can observe that the low-frequency information (e.g., brightness and color information) have been removed while the mid-frequency and high-frequency information (e.g., shapes and outlines) are kept intact.
> >
> > - In spatial masking, the un-masked pixels retain the original pixel values. In spectral masking, removing any image spectra will generally change the value of all image pixels as image spectra (also called image frequency bands) capture information of each and every pixel globally.
> >
> > - As mentioned in Section 3.2 (Lines159-168), the spectral masking can  encourage the networks to learn from all image spectral bands instead of focusing on easy bands (e.g., low-frequency bands capturing color and brightness) only.
> >
> > - Besides, both the Fourier transformation and pixel zeroing operation are differentiable, which will not leak the augmented information to the generator as proved in DA.

---

> ### Author Response · Authors · 2022-08-08
> **To Reviewer mM3P**
>
> Dear Reviewer mM3P:
>
> We thank you for the precious review time and valuable comments. We have provided corresponding responses and results, which we believe have covered your concerns. We hope to further discuss with you whether or not your concerns have been addressed. Please let us know if you still have any unclear parts of our work.
>
> Best regards,
> Authors

---

### Official Review · Reviewer_LGsj · 2022-07-12

**Rating:** 7
**Confidence:** 4
**Soundness:** 4 excellent
**Presentation:** 4 excellent
**Contribution:** 4 excellent

**Summary:**

This paper proposes two novel masked-based data augmentation methods for image generation task with GAN, namely shifted spatial masking and balanced spectral masking. Extensive experiments are conducted to demonstrate the effectiveness and generalizability of the proposed methods.

**Questions:**

1. Is the proposed method orthogonal to existing augmentation methods like [1][2]? The author might consider to add an experiment that using both the proposed mask-based augmentation with either [1] or [2] to see whether this would give an extra performance boost.

[1] Zhao, S., Liu, Z., Lin, J., Zhu, J. Y., & Han, S. (2020). Differentiable augmentation for data-efficient gan training. Advances in Neural Information Processing Systems, 33, 7559-7570.

[2] Karras, T., Aittala, M., Hellsten, J., Laine, S., Lehtinen, J., & Aila, T. (2020). Training generative adversarial networks with limited data. Advances in Neural Information Processing Systems, 33, 12104-12114.

**Ethics Review Area:**

["I don’t know"]

**Limitations:**

The author may consider to put more visualization results in the paper.

**Strengths And Weaknesses:**

Pro:
1. The paper is well-written and easy to follow.
2. The proposed method is technically sound and intuitive. The paper is well-motivated.
3. Extensive experiments are conducted to support the author's points.
4. Competitive performances are achieved by using the proposed method. The proposed method is generalizable enough to a lot of GAN architectures.

Cons:
1. Missing augmentation comparison: in table 3, I think a proper comparison would be 1. biggan, 2. biggan + diffaug[1], 3. biggan + [2], 4. biggan + masked-aug (MaskedGAN). Same issue also exists in table. 5~7.

[1] Zhao, S., Liu, Z., Lin, J., Zhu, J. Y., & Han, S. (2020). Differentiable augmentation for data-efficient gan training. Advances in Neural Information Processing Systems, 33, 7559-7570.

[2] Karras, T., Aittala, M., Hellsten, J., Laine, S., Lehtinen, J., & Aila, T. (2020). Training generative adversarial networks with limited data. Advances in Neural Information Processing Systems, 33, 12104-12114.

---

> ### Author Response · Authors · 2022-08-02
> **Thank you for your appreciation of our work. Below please find our detailed clarification for your raised questions. Part 1 (Q1)**
>
> Q1: Missing augmentation comparison? In table 3, a proper comparison would be 1. biggan, 2. biggan + diffaug[1], 3. biggan + [2], 4. biggan + masked-aug (MaskedGAN)?
> - Thank you for the suggestion! We conducted the suggested experiments over ImageNet in the revised Table 3. It can be seen that the proposed MaskedGAN outperforms other data augmentation methods (i.e., ADA and DA) clearly, which is largely attributed to our two masking designs. The experiments are consistent with other experiments in revised Tables 1 and 4 (including several newly conducted experiments), which together show that MaskedGAN outperforms other data augmentation methods consistently over various datasets (e.g., CIFAR-10, CIFAR-100, ImageNet and 100-shot datasets) and backbones (e.g., BigGAN and StyleGAN-v2). Thank you for your suggestion and we will include these comparisons in the revised appendix/manuscript.
> - Thank you for the suggestion again! Due to the time and computation resource constraints, we only benchmarked DA and ADA over the very large ImageNet dataset, the very small 100-shot dataset and the medium-size CIFAR-10/100 dataset, where we believe extensive experiments over these datasets are sufficient for benchmarking the proposed method. We plan to conduct related experiments and benchmark DA and ADA over other datasets and backbones later.
>
> Revised Table 3: Conditional image generation with BigGAN on ImageNet (FID).
> | Method                          | 10% Data | 5% Data  | 2.5% Data |
> | --------------------------------|:-------:|:-----:|:-----:|
> |BigGAN (baseline)    |38.30  ±  0.25   |91.16  ±  0.43  |133.80  ±  0.76
> |ADA (**newly included**)   | 31.89  ±0.17 |43.21   ±  0.37  | 56.83   ±  0.48
> |DA (**newly included**)    | 32.82    ±  0.18  |56.75   ±  0.35 |63.49   ±  0.51
> |**MaskedGAN**   | 26.51  ±  0.12  | 35.70  ±  0.31   | 38.62  ±  0.37
>
> Revised Table 3: Conditional image generation with BigGAN on ImageNet (IS).
> | Method                          | 10% Data | 5% Data  | 2.5% Data |
> | --------------------------------|:-------:|:-----:|:-----:|
> |BigGAN (baseline)  |10.94  ±  0.35 |6.13  ±  0.09  |3.92  ±  0.07  |
> |ADA (**newly included**)  |12.67  ±0.31   |9.44  ±0.25   |8.54  ±   0.26  |
> |DA (**newly included**)  |12.76  ±  0.34    |9.63  ±  0.21  |8.17  ±  0.28 |
> |**MaskedGAN**   |13.34  ±  0.24   | 12.85  ±  0.40  | 12.68  ±  0.27  |
>
> Revised Table 1: Conditional image generation with BigGAN on CIFAR-10 (FID).
> | Method                          | 100% Data | 20% Data  | 10% Data |
> | --------------------------------|:-------:|:-----:|:-----:|
> | Non-saturated GAN |9.83  ±  0.06  |18.59 ±  0.15   |41.99 ±  0.18 |
> | LS-GAN |9.07 ±  0.01 | 21.60 ±  0.11 | 41.68 ±  0.18
> | RAHinge GAN | 11.31 ±  0.04 | 23.90 ±  0.22| 48.13 ±  0.33|
> | StyleGAN-v2 | 11.07 ±  0.03  | 23.08 ±  0.11  | 36.02 ±  0.15  |
> | BigGAN (baseline)  | 9.74 ±  0.06| 21.86 ±  0.29| 48.08 ±  0.10 |
> | LeCam-GAN| 8.31 ±  0.05 |15.27 ±  0.10 |35.23 ±  0.14 |
> | GenCo | 8.83 ±  0.04 | 16.57 ±  0.08 |  28.08 ±  0.11|
> | ADA (**newly included**) |8.99  ±  0.03 |19.87  ±  0.09 |30.58  ±  0.11 |
> | DA | 8.75 ±  0.03 |  14.53 ±  0.10| 23.34 ±  0.09 |
> | **MaskedGAN**  | 8.41 ±  0.03 | 12.51 ±  0.09| 15.89  ±  0.12 |
>
> Revised Table 1: Conditional image generation with BigGAN on CIFAR-100 (FID).
> | Method                          | 100% Data | 20% Data  | 10% Data |
> | --------------------------------|:-------:|:-----:|:-----:|
> | Non-saturated GAN | 13.87 ±  0.08|  32.64  ±  0.19 |  70.5 ±  0.38|
> | LS-GAN|  12.43 ±  0.11|  27.09 ±  0.09|  54.69 ±  0.12  |
> | RAHinge GAN| 14.61 ±  0.21| 28.79 ±  0.17|  52.72 ±  0.18|
> | StyleGAN-v2| 16.54 ±  0.04 |  32.30  ±  0.11|  45.87 ±  0.15 |
> | BigGAN (baseline) | 13.60 ±  0.07|  32.99 ±  0.24| 66.71 ±  0.01|
> | LeCam-GAN | 11.88 ±  0.12| 25.51 ±  0.19| 49.63 ±  0.16|
> | GenCo|   11.90 ±  0.02| 26.15} ±  0.08| 40.98 ±  0.09|
> | ADA (**newly included**) | 12.22  ±  0.02|  22.65  ±  0.10| 27.08  ±  0.15|
> | DA|   11.99 ±  0.02|22.55 ±  0.06|35.39 ±  0.08 |
> | **MaskedGAN** |   11.65 ±  0.03|  18.33 ±  0.09| 24.02 ±  0.12|
>
> Revised Table 4: Unconditional image generation with StyleGAN-v2 on 100-shot dataset (FID).
> | Method                          | Obama | Grumpy Cat  | Panda |
> | --------------------------------|:-------:|:-----:|:-----:|
> Scale/shift  | 50.72  | 34.20  | 21.38
> MineGAN  | 50.63  |34.54  |14.84
> TransferGAN   |48.73  | 34.06  |23.20
> TransferGAN + DA   |39.85  |29.77 | 17.12
> FreezeD  | 41.87  | 31.22 | 17.95
> StyleGAN-v2  (baseline) | 80.20  | 48.90  | 34.27
> ADA  | 45.69  | 26.62 | 12.90
> LeCam-GAN  | 38.58  | 41.38  | 19.88
> GenCo  | 36.35  | 33.57 |  15.50
> AdvAug  | 52.86  | 31.02  | 14.75
> DA  |46.87 | 27.08 | 12.06
> APA (**newly included**)  | 43.75  | 28.49  |  12.34
> InsGen (**newly included**)  | 45.85  | 27.48  | 12.13
> **MaskedGAN**   | 33.78 ± 0.27  | 20.06 ± 0.13  | 8.93 ± 0.06

---

> > ### Author Response · Authors · 2022-08-02
> > **Part 2 (Q2-Q3)**
> >
> > Q2: Is the proposed method orthogonal to existing augmentation methods like DA [1] ADA [2]?
> > - We did not consider incorporating DA and ADA into MaskedGAN because their data augmentation strategies violate the proposition condition of removing certain image information only (Propositions 1 and 2 provided in Section 3.3). In other words, incorporating DA/ADA into MaskedGAN may impair the theoretic convergence guarantee of MaskedGAN and even deteriorate the training stability and generation performance. Nevertheless, it is interesting to explore how and whether MaskedGAN and DA/ADA complement and we will definitely investigate this issue later.
> >
> > Q3: The author may consider to put more visualization results in the paper.
> > - Thank you for the suggestion! We will include more visualization results in the revised version.

---

### Official Review · Reviewer_aPDi · 2022-07-13

**Rating:** 5
**Confidence:** 4
**Soundness:** 2 fair
**Presentation:** 3 good
**Contribution:** 2 fair

**Summary:**

This paper propses MaskedGAN to help GANs learn from the limited data by introducing two image masking strategies. One is the shifted spatial masking and the other is the balanced spectral masking. Those two masking strategies complement each other which together encourage GANs learning effectively and robustely on the limited data.

**Questions:**

1. Differences from data augmentation? As it lacks the reconstruction part of MAE, the masking strategy can be regarded as another type of data augmentation. From this perspective, what if we have sufficient data, is improvement consistent? Besides, the relation to data augmentation is required to be discussed in related work.

2. As claimed in the contribution summary, the holistic understanding of images is encouraged. Is there any experimental support?

**Limitations:**

The authors do not adequately addressed the limitations and potential negative societal impact of their work.

**Strengths And Weaknesses:**

Strengths:

The motivation of this paper is clear, and the authors did lots of experiments to demonstrate the effectiveness of their method. But the experiment setting part needs to be improved as listed in the weaknesses or question part.


Weaknesses:

1. More data augmentation methods are supposed to be involved in the comparison of CIFAR and Imagenet. For example, StyleGAN2-ADA is required rather than the original StyleGAN2, especially with limited training data. Overall, some tables seem unfair since many baselines are without data augmentations, while the proposed method could be regarded as one novel data augmentation.

2. Some important baselines are missing. For instance, InsGen and APA (both are NeurIPS 2021 work) achieved strong performances with limited training data on FFHQ, and AFHQ.

InsGen: Data-Efficient Instance Generation from Instance Discrimination, Yang. et al, NeurIPS 2021.
APA: Deceive D: Adaptive Pseudo Augmentation for GAN Training with Limited Data, Jiang, et al. NeurIPS 2021.

---

> ### Author Response · Authors · 2022-08-02
> **Thank you for your acknowledgment of the clear motivation and impressive experimental results. Below please find our responses regarding your concerns Part 1 (Q1-Q2).**
>
> Q1: More data augmentation methods are supposed to be involved in the comparison of CIFAR and ImageNet?
> - Thank you for the suggestion! We compared with the suggested two state-of-the-art methods DA and ADA that achieve data-limited generation through data augmentation. The proposed MaskedGAN achieves outstanding performance consistently across multiple datasets (e.g., CIFAR-10, CIFAR-100, ImageNet and 100-shot datasets) and backbones (e.g., BigGAN and StyleGAN-v2) as shown in revised Tables 1, 3, and 4 (newly conducted experiments are highlighted). The outstanding performance is largely attributed to our two masking designs (i.e., shifted spatial masking and balanced spectral masking) which randomly remove certain image information during network training and thus encourage a holistic learning of images as illustrated in Section A in appendix. Thank you for your suggestion and we will include the new experiments in our manuscript.
>
> Revised Table 1: Conditional image generation with BigGAN on CIFAR-10 (FID).
> | Method                          | 100% Data | 20% Data  | 10% Data |
> | --------------------------------|:-------:|:-----:|:-----:|
> | Non-saturated GAN |9.83  ±  0.06  |18.59 ±  0.15   |41.99 ±  0.18 |
> | LS-GAN |9.07 ±  0.01 | 21.60 ±  0.11 | 41.68 ±  0.18
> | RAHinge GAN | 11.31 ±  0.04 | 23.90 ±  0.22| 48.13 ±  0.33|
> | StyleGAN-v2 | 11.07 ±  0.03  | 23.08 ±  0.11  | 36.02 ±  0.15  |
> | BigGAN (baseline)  | 9.74 ±  0.06| 21.86 ±  0.29| 48.08 ±  0.10 |
> | LeCam-GAN| 8.31 ±  0.05 |15.27 ±  0.10 |35.23 ±  0.14 |
> | GenCo | 8.83 ±  0.04 | 16.57 ±  0.08 |  28.08 ±  0.11|
> | ADA (**newly included**) |8.99  ±  0.03 |19.87  ±  0.09 |30.58  ±  0.11 |
> | DA | 8.75 ±  0.03 |  14.53 ±  0.10| 23.34 ±  0.09 |
> | **MaskedGAN**  | 8.41 ±  0.03 | 12.51 ±  0.09| 15.89  ±  0.12 |
>
> Revised Table 1: Conditional image generation with BigGAN on CIFAR-100 (FID).
> | Method                          | 100% Data | 20% Data  | 10% Data |
> | --------------------------------|:-------:|:-----:|:-----:|
> | Non-saturated GAN | 13.87 ±  0.08|  32.64  ±  0.19 |  70.5 ±  0.38|
> | LS-GAN|  12.43 ±  0.11|  27.09 ±  0.09|  54.69 ±  0.12  |
> | RAHinge GAN| 14.61 ±  0.21| 28.79 ±  0.17|  52.72 ±  0.18|
> | StyleGAN-v2| 16.54 ±  0.04 |  32.30  ±  0.11|  45.87 ±  0.15 |
> | BigGAN (baseline) | 13.60 ±  0.07|  32.99 ±  0.24| 66.71 ±  0.01|
> | LeCam-GAN | 11.88 ±  0.12| 25.51 ±  0.19| 49.63 ±  0.16|
> | GenCo|   11.90 ±  0.02| 26.15} ±  0.08| 40.98 ±  0.09|
> | ADA (**newly included**) | 12.22  ±  0.02|  22.65  ±  0.10| 27.08  ±  0.15|
> | DA|   11.99 ±  0.02|22.55 ±  0.06|35.39 ±  0.08 |
> | **MaskedGAN** |   11.65 ±  0.03|  18.33 ±  0.09| 24.02 ±  0.12|
>
>
> Revised Table 3: Conditional image generation with BigGAN on ImageNet (FID).
> | Method                          | 10% Data | 5% Data  | 2.5% Data |
> | --------------------------------|:-------:|:-----:|:-----:|
> |BigGAN (baseline)    |38.30  ±  0.25   |91.16  ±  0.43  |133.80  ±  0.76
> |ADA (**newly included**)   | 31.89  ±0.17 |43.21   ±  0.37  | 56.83   ±  0.48
> |DA (**newly included**)    | 32.82    ±  0.18  |56.75   ±  0.35 |63.49   ±  0.51
> |**MaskedGAN**   | 26.51  ±  0.12  | 35.70  ±  0.31   | 38.62  ±  0.37
>
> Revised Table 3: Conditional image generation with BigGAN on ImageNet (IS).
> | Method                          | 10% Data | 5% Data  | 2.5% Data |
> | --------------------------------|:-------:|:-----:|:-----:|
> |BigGAN (baseline)  |10.94  ±  0.35 |6.13  ±  0.09  |3.92  ±  0.07  |
> |ADA (**newly included**)  |12.67  ±0.31   |9.44  ±0.25   |8.54  ±   0.26  |
> |DA (**newly included**)  |12.76  ±  0.34    |9.63  ±  0.21  |8.17  ±  0.28 |
> |**MaskedGAN**   |13.34  ±  0.24   | 12.85  ±  0.40  | 12.68  ±  0.27  |
>
> Q2: Overall, some tables seem unfair since many baselines are without data augmentations, while the proposed method could be regarded as one novel data augmentation?
> - Data-limited image generation has been tackled in two typical approaches, namely, data augmentation approach like DA and ADA and model regularization approach like LeCAM-GAN and GenCo. The two approaches address data constraint from very different perspectives and they are quite independent, i.e., data augmentation approach usually won't involve model regularization and vice versa. That's why, unlike the benchmarking with data augmentation baselines like DA and ADA, several model regularization baselines involve little data augmentation in benchmarking. Note such evaluation practice has been widely adopted in the data-limited image generation studies such as LeCAM-GAN and GenCo.

---

> > ### Author Response · Authors · 2022-08-02
> > **Part 2 (Q3)**
> >
> > Q3: Some important baselines are missing?
> > - Thank you for the suggestion! We compared with the suggested InsGen and APA. The experiments show that the proposed MaskedGAN outperforms the two methods clearly (as shown in the revised Table 4), which is largely attributed to our two masking designs (i.e., shifted spatial masking and balanced spectral masking) which randomly remove certain image information during network training and thus encourage a holistic learning of images as illustrated in Section A in appendix. Thank you for your suggestion and we will include the new experiments in our manuscript.
> > - We would clarify that we did not benchmark with ADA-related methods including InsGen and APA as our work mainly follows DA that adopts different experiment setups. To benchmark with ADA-related methods, it requires to rerun their codes over DA's experiment setups for valid comparisons. Such experiments are super computational intensive, e.g., ADA takes approximately 1, 259, 337.6 GPU hours (i.e., 143.76 GPU years) to complete its experiments. We did not have sufficient GPU resources to conduct such experiments.
> > - Therefore, due to the computation resource constraints, we only benchmarked InsGen and APA over 100-shot dataset. Nevertheless, the experiments and comparisons over the 100-shot dataset (i.e., the revised Table 4) are very relevant and meaningful as this paper focuses on training GAN with limited data, where the task would be more challenging while working with a small dataset.
> >
> > Revised Table 4: Unconditional image generation with StyleGAN-v2 on 100-shot dataset (FID).
> > | Method                          | Obama | Grumpy Cat  | Panda |
> > | --------------------------------|:-------:|:-----:|:-----:|
> > Scale/shift  | 50.72  | 34.20  | 21.38
> > MineGAN  | 50.63  |34.54  |14.84
> > TransferGAN   |48.73  | 34.06  |23.20
> > TransferGAN + DA   |39.85  |29.77 | 17.12
> > FreezeD  | 41.87  | 31.22 | 17.95
> > StyleGAN-v2 (baseline)  | 80.20  | 48.90  | 34.27
> > ADA  | 45.69  | 26.62 | 12.90
> > LeCam-GAN  | 38.58  | 41.38  | 19.88
> > GenCo  | 36.35  | 33.57 |  15.50
> > AdvAug  | 52.86  | 31.02  | 14.75
> > DA  |46.87 | 27.08 | 12.06
> > APA (**newly included**)  | 43.75  | 28.49  |  12.34
> > InsGen (**newly included**)  | 45.85  | 27.48  | 12.13
> > **MaskedGAN**   | 33.78 ± 0.27  | 20.06 ± 0.13  | 8.93 ± 0.06

---

> > > ### Author Response · Authors · 2022-08-02
> > > **Part 3 (Q4-Q5)**
> > >
> > > Q4: Differences from data augmentation?
> > > - We would clarify that we discussed the difference between our MaskedGAN and data augmentation methods extensively in our manuscript and appendix. For example, the 4th paragraph of Section 4.4 describes the better convergence of the proposed MaskedGAN and why it converges better than existing data augmentation methods; Section 3.3 provides detailed theoretical insights and illustrations (on two time-scale update rule and local Nash Equilibrium); Section H of appendix shows the comparison with "cutout" that is used in data augmentation methods DA and ADA.
> > > - In summary, MaskedGAN designs two image masking strategies that work by removing certain image information only while previous data augmentation methods involve various data augmentations such as color jitters, saturation adjustment, etc. Besides, different to the conventional "cutout’ in DA and ADA, MaskedGAN uses "random patch-based masking" and designs "random mask shift" and "balanced spectral masking".
> > > Such differences in designs lead to very different results in convergences (empirically and theoretically) and generation performances.
> > > More detailed descriptions can be found in the following texts. Thank you for your suggestion and we will include the above discussions in the revised manuscript.
> > >
> > >
> > > - Followed please find the detailed texts copied (or summarized) from our manuscript and appendix for your reference, which extensively discuss the difference between our MaskedGAN and data augmentation methods:
> > >
> > > - **1)** As mentioned in the 4th paragraph ("Convergence comparison across different network architectures and datasets") of Subsection 4.4, the experimental results show that MaskedGAN converges well consistently across various conditions (the amounts of training data, network architectures and datasets) while the data augmentation method such DA still suffers from generation failures and training collapses. The great convergence of MaskedGAN is largely attributed to two factors: (a) its image masking designs suppress trivial solutions and training failures directly; (b) it keeps similar learning paces for discriminator and generator which ensures that network converge to a Local Nash Equilibrium under certain conditions [21].
> > > MaskedGAN can achieve factor (b) as its image masking strategies (or called data augmentation operations) operate by masking (or more specifically, removing) some image information only.
> > > Differently, data augmentation methods such as DA and ADA cannot guarantee factor (b) as they generally includes some operations like color jitters and saturation adjustment which cannot satisfy the theoretic proofs as introduced in Section 3.3.
> > >
> > > - **2)** Point 1) described the design differences between MaskedGAN and previous data augmentation methods (e.g., DA and ADA) and illustrated different designs lead to different empirical convergences and further different generation performances.
> > > In Section 3.3, we provided detailed theoretic insights and illustrations, which show MaskedGAN can be modeled as an instance of Two Time-Scale Update Rule and and thus it converges to a Local Nash Equilibrium under certain conditions.
> > > On the other hand, previous data augmentation methods (e.g., DA and ADA) cannot satisfy our Propositions 1 and 2 provided in Section 3.3 because data augmentation methods such as DA and ADA includes some operations like color jitters and saturation adjustment, where such additive noises violate the proposition condition of removing certain image information only.
> > >
> > > - **3)** In Section H in appendix ("Comparisons with the ‘cutout’ used in ADA and DA"), we compared our MaskedGAN with the ‘cutout’ used in ADA and DA by providing experiments and detailed explanations.
> > >
> > > Q5: As claimed in the contribution summary, the holistic understanding of images is encouraged. Is there any experimental support?
> > > - This is really a critical issue, and we attempted to address it with the Gini coefficient of spatial attention in Section A of the appendix. As Fig. 1 (in Section A of the appendix) shows, the baseline model tends to merely focus on a few image locations without holistic understandings of images, ultimately leading to an over-confident discriminator and training collapse. In contrast, MaskedGAN pays very even attention to every spatial location (i.e., learning and understanding images more holistically), resulting in more stable training process and better performance. More details can be found in Section A in appendix.

---

> > > > ### Comment · Reviewer_aPDi · 2022-08-09
> > > > **Update**
> > > >
> > > > Thanks for the timely and detailed response from the authors. My concerns have been addressed, and I will raise the score.

---

> ### Author Response · Authors · 2022-08-08
> **To Reviewer aPDi:**
>
> Dear Reviewer aPDi:
>
> We thank you for the precious review time and valuable comments. We have provided corresponding responses and results, which we believe have covered your concerns. We hope to further discuss with you whether or not your concerns have been addressed. Please let us know if you still have any unclear parts of our work.
>
> Best regards,
> Authors

---

### Meta-Review · Area_Chair_zffH · 2022-08-24

**Recommendation:** Accept
**Confidence:** Less certain

**Metareview:**

This paper proposes two masking strategies to improve GANs with limited data. The idea is novel and these two strategies can nicely complement each other. The experiment results are promising. The reviewers unanimously raised questions on missing comparison, which seem to be well addressed after author-reviewer discussion. Two reviewers end up raising their scores. There are still some claims in the paper that may need better experimental evidence support, and also more visual cues would be helpful in paper presentation. Also, all reviewers pointed out that the discussion of limitations and broader impact seem not adequate. I would recommend weak acceptance however strongly encourage the authors to address the above-mentioned concerns in their next version.

**Award:**

No

---

### Decision · Program_Chairs · 2022-09-14

Accept